# RETHINK MAXIMUM STATE ENTROPY

## ABSTRACT

In the absence of specific tasks or extrinsic reward signals, a key objective for an agent is the efficient exploration of its environment. A widely adopted strategy to achieve this is maximizing state entropy, which encourages the agent to uniformly explore the entire state space. Most existing approaches for maximum state entropy (MaxEnt) are rooted in two foundational approaches, which were proposed by Hazan and Liu & Abbeel, respectively. However, a unified perspective on these methods is lacking within the community.

In this paper, we analyze these two foundational approaches within a unified framework and demonstrate that both methods share the same reward function when employing the $k$NN density estimator. We also show that the $\eta$-based policy sampling method proposed by Hazan is unnecessary and that the primary distinction between the two lies in the frequency with which the locally stationary reward function is updated. Building on this analysis, we introduce MaxEnt-(V)eritas, which combines the most effective components of both methods: iteratively updating the reward function as defined by Liu & Abbeel, and training the agent until convergence before updating the reward functions, akin to the procedure used by Hazan. We prove that MaxEnt-V is an efficient $\varepsilon$-optimal algorithm for maximizing state entropy, where the tolerance $\varepsilon$ decreases as the number of iterations increases. Empirical validation in three Mujoco environments shows that MaxEnt-Veritas significantly outperforms the two MaxEnt frameworks in terms of both state coverage and state entropy maximization, with sound explanations for these results.

## 1 INTRODUCTION

Reinforcement Learning (RL) has demonstrated remarkable success in domains such as robotics (Mnih et al., 2015) and games (Silver et al., 2016). Nevertheless, a fundamental challenge in RL is the effective exploration of the state space in the absence of extrinsic reward signals. Recently, state entropy $H(\mathbf{s})$ has emerged as a robust metric for quantifying the diversity of state coverage, thereby making the maximum state entropy (MaxEnt) framework a widely adopted paradigm for exploration (Liu & Abbeel, 2021; Mutti et al., 2021; Seo et al., 2021; Yuan et al., 2023; Hazan et al., 2019; Zhang et al., 2021; Nedergaard & Cook, 2022; Yarats et al., 2021; Tiapkin et al., 2023; Kim et al., 2024). The principal objective of the MaxEnt framework is to derive a policy that facilitates uniform exploration of all possible states.

Most existing approaches to state entropy maximization are grounded in two foundational works: the first, proposed by Hazan (Hazan et al., 2019) (MaxEnt-H), introduces a provably efficient $\varepsilon$-optimal algorithm for maximizing the entropy of visited states, assuming access to (sub)-optimal planning policies (e.g., by training deep reinforcement learning agents to convergence). Building on this foundation, subsequent work has focused on reducing computational complexity (Tiapkin et al., 2023), extending the approach to Rényi entropy (Zhang et al., 2021), and other advancements (Nedergaard & Cook, 2022; Yarats et al., 2021). While these importance sampling-based methods have made significant theoretical contributions, they operate under the assumption that we can "compute the (approximately) optimal policy" to solve a MDP at each iteration given the locally stationary reward function. This assumption is often unrealistic in non-tabular settings. The other type, introduced by Liu & Abbeel (2021) (MaxEnt-LA), decomposes $k$-nearest neighbor ($k$NN) entropy estimation into "particles" and uses these as non-stationary dense rewards to train a deep reinforcement learning (DRL) agent. These $k$NN-based methods (Singh et al., 2003) have been widely applied to improve

sample efficiency, facilitate unsupervised pre-training for downstream tasks, and more. Although lacking theoretical guarantees, MaxEnt-LA and its subsequent developments achieve state-of-the-art performance in complex environments (Liu & Abbeel, 2021; Seo et al., 2021; Yuan et al., 2023; Kim et al., 2024). Given the prominence of these two methods and their following variants, a natural question arises: Is there a connection between them for exploration, particularly in the absence of extrinsic rewards? In this paper, we provide an explicit answer to this question.

---

**Algorithm 1** Pipeline of MaxEnt frameworks. Blue text represents steps specific to MaxEnt-H, while red text corresponds to steps for MaxEnt-LA.

---

**Require**: Step size $\eta$ and the set of sampling probability $A_0 = \{\alpha_0\}$. Initialize RL agent as $\pi_0$.

1: **for** $t = 0, 1 \cdots T - 1$ **do**
2:    MaxEnt-H samples $\{\pi_0, \pi_1 \cdots \pi_t\}$ with probability $\{\alpha_0, \alpha_1 \cdots \alpha_t\}$ to induce states.
      MaxEnt-LA samples $\{\pi_0, \pi_1 \cdots \pi_t\}$ uniformly to induce states.
3:    Define intrinsic reward functions $r_t^H(s)$ or $r_t^{LA}(s)$ based on states induced by $\{\pi_0, \pi_1 \cdots \pi_t\}$.

4:    MaxEnt-H initializes $\pi$ and trains it with $r_t^H(s)$ until convergence to get $\pi_{t+1}$.
      MaxEnt-LA continues to train $\pi$ with $r_t^{LA}(s)$ for one step to get $\pi_{t+1}$.
5:    MaxEnt-H updates the set of sampling probabilities as $A_{t+1} = (1 - \eta)A_t \cup \{\alpha_{t+1} = \eta\}$.
6: **end for**
7: **return** $\{\pi_0, \pi_1 \cdots \pi_T\}$, $\{\alpha_0, \alpha_1 \cdots \alpha_T\}$.

---

We present a unified framework for both approaches in Algorithm 1: at each iteration, both MaxEnt frameworks (Hazan et al., 2019; Liu & Abbeel, 2021) begin by defining an intrinsic reward function based on the state distributions induced by previous policies, followed by training the current policy using this reward function. Subsequently, MaxEnt-H updates the sampling strategy for previous policies using a hyper-parameter $\eta$. Both methods then proceed to the next iteration. Three distinctions can be summarized as follows: (**D1**) They employ different methods for defining the reward function $r_t$ (Step 3). (**D2**) The policy sampling strategies diverge: MaxEnt-H utilizes an evolving distribution based on $\eta$, whereas MaxEnt-LA samples policies uniformly (Steps 2 and 5). (**D3**) The frequency of reward function updates during the training process also differs: MaxEnt-H trains the agent to convergence (or until a tolerance level is reached) before updating the reward function, while MaxEnt-LA updates the reward function after each individual training step (Step 4).

For **D1**, we prove that the reward function in MaxEnt-LA (Liu & Abbeel, 2021; Seo et al., 2021) is proportional to the reward function defined by Hazan when the $k$NN density estimator is employed. Concerning **D2**, we show that the $\eta$-based approach is superfluous for achieving a meaningful tolerance $\varepsilon$, especially in non-tabular state spaces; sampling previous policies randomly, as in MaxEnt-LA, is sufficiently effective. Consequently, the primary distinction lies in the frequency of reward function updates (**D3**). In this context, we argue that frequent updates to the reward function are suboptimal, as they cause the RL agent to continuously maximize a non-stationary reward function. Instead, the RL agent should be allowed to train until it performs satisfactorily, as evaluated by the current reward function, similar to the approach taken by MaxEnt-H.

Building on this rethinking, we propose that state maximization in non-tabular environments can be achieved with a highly simplified algorithm by integrating key elements from both approaches: **all you need is to iteratively update the reward function as defined in MaxEnt-LA and to train the RL agent until convergence (or until a predefined tolerance is reached), given the locally stationary reward function.** We refer to this method as **MaxEnt-(V)eritas**. Theoretically, we demonstrate that MaxEnt-V is a provably efficient $\varepsilon$-optimal algorithm for maximizing state entropy, where the tolerance $\varepsilon$ decreases at an approximate rate of $\frac{B + \beta \log T}{T}$, with $B$ and $\beta$ representing the bounds of the reward functional, which is assumed to be $\beta$-smooth and $B$-bounded. Empirically, we evaluate MaxEnt-V against the methods of Hazan et al. (2019) and Liu & Abbeel (2021) in the Mujoco robotic simulation environments, and it consistently outperforms both approaches in terms of state coverage and state entropy maximization. Our primary contributions are as follows:

- We elucidate the relationship between the two seminal MaxEnt frameworks proposed by Hazan and Liu&Abbeel. Specifically, we demonstrate that both approaches share an intrinsic reward function, that the $\eta$-based sampling method introduced by Hazan is redundant,

and that the principal distinction between the two lies in the frequency with which the reward function is updated.

- Building on the analysis, we introduce a novel intrinsically motivated policy learning method, termed MaxEnt-Veritas, which leverages the reward function proposed by Liu&Abbeel and sample policies randomly to facilitate pure exploration in non-tabular environments.

- MaxEnt-V is evaluated against the two MaxEnt frameworks across three exploration environments based on Mujoco. It consistently outperforms all competing approaches in experiments focused on exploring novel states.

## 2 PRELIMINARY

**Markov decision process:** an infinite-horizon Markov decision process (MDP) is defined by a 5-tuple $(\mathcal{S}, \mathcal{A}, P, r, \gamma)$, where $\mathcal{S}$ is the set of all possible states, $\mathcal{A}$ is the set of actions, $P(s_{i+1}|s_i, a_i) : \mathcal{S} \times \mathcal{A} \to \mathcal{S}$ is the transition probability density function. $\gamma \in [0, 1)$ is a discount factor. $r(s_i, a_i) : \mathcal{S} \times \mathcal{A} \to \mathbb{R}$ is a stationary reward function. The performance of an infinite trajectory $\tau$ of states and actions is judged through the (discounted) cumulative reward it accumulates, defined as $V(\tau = \{s_0, a_0, s_1, a_1 \cdots\}) = \sum_{i=0}^{\infty} \gamma^i [r(s_i, a_i)]$.

**Induced state distributions:** Given a policy $\pi(a|s) : \mathcal{S} \to \mathcal{A}$, the probability of the $\pi$-induced trajectory can be written as $P(\tau|\pi) = P(s_0) \prod_{i=0}^{\infty} \pi(a_i|s_i) P(s_{i+1}|s_i, a_i)$. The $i$-step state distribution and the (discounted) state distribution of $\pi$ are:

$$d_{\pi,i}(s) = P(s_i = s|\pi) = \sum_{\{\tau | s_i = s\}} P(\tau|\pi)$$

$$d_{\pi}(s) = \sum_{i=0}^{\infty} \gamma^i [d_{\pi,i}(s_i)] \tag{1}$$

The goal is to find an optimal policy $\pi^*$ that induces a state distribution with maximum entropy:

$$\pi^* = \arg \min_{\pi} H(d_{\pi}(s)) = \arg \min_{\pi} \left[ -\mathbb{E}_{s \in \mathcal{S}} \left( \log(d_{\pi}(s)) \right) \right] \tag{2}$$

In practice, we can execute policy $\pi$ from different initial states $s_0$ to sample a large number of states $s$. The estimated distribution $\hat{d}_{\pi}(s)$ can then be approximated by the empirical probability of these sampled states.

### 2.1 MAXENT BY HAZAN (MAXENT-H)

**Mixtures of stationary policies:** Given $k$ policies $C = \{\pi_0, \pi_1 \cdots \pi_{k-1}\}$, and corresponding sampling probabilities $A = \{\alpha_0, \alpha_1 \cdots \alpha_{k-1}\}$, MaxEnt-H defined $\pi_{\text{mix}} = (A, C)$ to be a mixture over these stationary policies. The (non-stationary) policy $\pi_{\text{mix}}$ is one where, at the first timestep $t = 0$, MaxEnt-H samples policy $\pi_i$ with probability $\alpha_i$ and then uses this policy for all subsequent time steps. The induced state distribution is:

$$d_{\pi_{\text{mix}}}(s) = \sum_{i=0}^{k-1} \alpha_i d_{\pi_i}(s) \tag{3}$$

While the entropy objective is not smooth, MaxEnt-H considers a smoothed alternative $H_{\sigma} = -\mathbb{E}_{s \sim d_{\pi}} \log(d_{\pi}(s) + \sigma)$. We shall assume in the following discussion that the reward functional $H_{\sigma}$ is $\beta$-smooth, $B$-bounded. The main theorem of MaxEnt-H for state entropy maximization is:

**Lemma 1** *(Hazan et al., 2019) We assumes that the RL agent in Algorithm 1 (blue) converges to an $\varepsilon_1$-optimal solution , given current reward function $r_t^H(s) = \nabla H(\hat{d}_{\pi_t}(s)) := \frac{dH(X)}{dX}\big|_{X = \hat{d}_{\pi_t}(s)}$. Meanwhile, we assume to guarantee the estimation error of state distribution $\|\hat{d}_{\pi_t}(s) - d_{\pi_t}(s)\|_{\infty} < \varepsilon_0$.*

*For any $\varepsilon > 0$, set $\sigma = \frac{0.1\varepsilon}{2|S|}$, $\varepsilon_1 = 0.1\varepsilon$, $\varepsilon_0 = \frac{0.1\varepsilon^2}{80|S|}$ and $\eta = \frac{0.1\varepsilon^2}{40|S|}$. When Algorithm 1 (blue) is run for T iterations with the reward functional $H_\sigma$, where:*

$$T \geq \frac{40|S|}{0.1\varepsilon^2} \log \frac{\log |S|}{0.1\varepsilon}, \tag{4}$$

*we have that:*

$$H_\sigma(d_{\pi_{mix},T}) \geq \max_\pi H_\sigma(d_\pi) - \varepsilon \tag{5}$$

## 2.2 MaxEnt by Liu & Abbeel (MaxEnt-LA)

MaxEnt-LA does not formulate the problem as a traditional MDP with a stationary reward function. Instead, it seeks to directly replace extrinsic rewards with decomposed $k$NN entropy estimates over time, which are inherently non-stationary. Let $s_i^{kNN}$ be the $k$NNs of $\mathbf{s}_i$, the $k$NN entropy estimate $H_{kNN}$ is given by (Singh et al., 2003):

$$H_{kNN}(d(s)) = \frac{1}{N} \sum_{i=1}^{N} \log \frac{N \cdot ||s_i - s_i^{kNN}||_2^p \cdot \pi^{p/2}}{k \cdot \Gamma(p/2 + 1)} + C_k \propto \frac{1}{N} \sum_{i=1}^{N} \log ||s_i - s_i^{kNN}||_2^p, \tag{6}$$

where $C_k = \log k - \Psi(k)$ is a bias correction constant, in which $\Psi$ is the digamma function; $\Gamma$ is the gamma function; $p$ is the dimentionality of $s$. The $r_t^{LA}(s)$ is defined as:

$$r^{LA}(s) = \log(||s - s^{kNN}||_2^p) \tag{7}$$

Notice that, $s^{kNN}$ is computed using all historical states. Such a reward function is not be representable as a stationary aim due to that the $||s - s^{kNN}||_2$ are no longer conditionally independent given the states.

## 3 Analysis of MaxEnt Frameworks

As illustrated in Algorithm 1, the pipeline of both MaxEnt frameworks can be described as iteratively updating the non-stationary intrinsic reward function $r_t(s)$ and training an agent to maximize the (discounted) accumulated rewards based on this function. Based on the comparison, the key differences can be summarized as follows:

- **(D1)** The reward functions, $r_t^{LA}$ and $r_t^H$ (Step 3).
- **(D2)** The method for sampling policies (Steps 2 and 5). MaxEnt-V samples previous policies using a dynamic $\eta$-based distribution, while MaxEnt-LA samples them uniformly.
- **(D3)** The frequency of reward function updates during the training process (Step 4). MaxEnt-V trains the agent until an $\varepsilon_1$-optimal solution is achieved for $r_t^H$ in each iteration, whereas MaxEnt-LA updates the agent's parameters for a single step in each iteration.

In this section, we will discuss each of these points in detail. We begin by examining the definition of the reward functions, as follows:

**Proposition 1** *Assuming the use of the kNN density estimator to approximate $d_\pi(s)$ in a state distribution, we have $r^H(s) \propto r^{LA}(s)$.*

Given the definition $r_t^H(s) = \nabla H(\hat{d}_{\pi_t}(s)) := \frac{dH(X)}{dX}|_{X=\hat{d}_{\pi_t}(s)}$, we have:

$$r_t^H(s) = \log(\frac{1}{\hat{d}_{\pi_t}(s)}) - 1 \tag{8}$$

when we adopt $k$NN density estimator, we have:

$$\hat{d}_{\pi_t}(s) = \frac{k \cdot \Gamma(p/2 + 1)}{N \cdot ||s_i - s_i^{kNN}||_2^p \cdot \pi^{p/2}} \tag{9}$$

Then,

$$r_t^H(s) = \log(\frac{N \cdot ||s_i - s_i^{kNN}||_2^p \cdot \pi^{p/2}}{k \cdot \Gamma(p/2 + 1)}) - 1 \tag{10}$$

Recall that $r^{LA}(s) = \log(\|s - s^{kNN}\|_2^p)$, we have $r^H(s) \propto r^{LA}(s)$. In short, $r^H$ and $r^{LA}$ can be regarded as equivalent during the training process if probability values are estimated using $k$NNs. We now move on to the second difference. With respect to the policy sampling method in MaxEnt-H, it is essential to consider the maximum possible state entropy value, i.e., $\max H(s) = \log |S|$, in Lemma 1. Notice that, however, the maximum state entropy value is not strictly $\log |S|$ due to the influence of the smoothness factor $\sigma$ in MaxEnt-H. In this paper, we omit further discussion of this aspect given the tiny magnitude of $\sigma$. In this context, we have that:

**Proposition 2** *When Algorithm 1 (blue) is performed, the step size must satisfy $\eta < \frac{\log^2(|S|)}{400|S|}$ in order to achieve any tolerance $\varepsilon < \log |S|$. For any $|S| \geq 2$, this implies a step size $\eta < 0.00136$ to ensure a tolerance $\varepsilon < \log |S|$.*

Considering the small value of $\eta$, the MaxEnt-H sampling method essentially behaves as uniform sampling when $T$ is not significantly large (recall that probability $\alpha_t = \eta(1 - \eta)^{t-1}$ when $t > 0$). However, in practice, the $T$ of MaxEnt-H cannot be substantial, as each iteration corresponds not to a single training step, but rather to training the agent until convergence. Another critical issue is that the probability $\alpha_t = \eta(1 - \eta)^{t-1}$, with a fixed $\eta$, can never satisfy the condition $\sum_{t=0}^{T} \alpha_t = 1$. In the MaxEnt-H paper, the authors addressed this issue by setting $\alpha_0 = 1$ prior to iteration 1. However, this was not implemented in their experiments, as it would cause the agent to select $\pi_0$ (random action selection) most of the time if $T$ is small. Instead, they attempt to solve an optimization problem subject to the constraint $\sum_{t=0}^{T} \alpha_t = 1$, subsequently normalizing the probabilities by dividing by the sum $\sum_{t=0}^{T} \alpha_t$. Further details can be found in Section 5.1. In this context, $\eta$ becomes a dynamic value and thus contradicts the theoretical framework established by MaxEnt-H. Consequently, we argue that $\eta$-based sampling is redundant in the context of MaxEnt framework.

Given the analysis of **D1** and **D2**, the only substantial difference between MaxEnt-H and MaxEnt-LA lies in the frequency of reward function updates. We argue that it is preferable to train the RL agent until convergence (as in MaxEnt-H), rather than after each individual step (as in MaxEnt-LA), before updating the reward function. This approach intuitively improves the stationarity of the reward function (or goal) that the agent seeks to maximize. If the reward function is updated after every training step, the agent will be encouraged to learn different reward functions at each step.

Unfortunately, training agents to convergence in each iteration naturally limits $T$ to a small value in practice, as it is infeasible to train millions of DRL agents. This leads to another gap between theoretical guarantees given by MaxEnt-H and real-world applications:

**Proposition 3** *When Algorithm 1 (blue) is run for $T$ iterations, for any tolerance $\varepsilon < \log |S|$, we have the number of iterations $T > \frac{1329|S|}{\log^2 |S|}$.*

This proposition demonstrates that MaxEnt requires a large $T$ to provide a meaningful guarantee $\varepsilon < \log |S|$. For instance, if $|S| = 10$, MaxEnt-H would require approximately $T = 4,000$ iterations to guarantee $\varepsilon \approx \log |S|$, which essentially corresponds to no policy improvement. In the experimental implementation of MaxEnt-H, the state spaces $S$ were discretized into $64$ to $194,400,000$ bins, depending on the environment, yet $T$ was set to a maximum of $30$.

In summary, the reward functions can be regarded as equivalent, and the $\eta$-based sampling method should be dismissed in favor of simplicity, in line with Occam's Razor. Meanwhile, we should train the agent until convergence using the locally stationary reward functions. Unfortunately, theoretical guarantees given by MaxEnt-H are not practical for guiding empirical approaches in non-tabular environments. This raises a natural question: Can we develop a novel MaxEnt algorithm with meaningful guarantees within a reasonable number of iterations? We will address this in the next section.

## 4 TRUE MAXIMUM STATE ENTROPY (MAXENT-VERITAS)

We propose MaxEnt-Veritas, a streamlined approach that peels MaxEnt algorithms to the bone by eliminating all redundancies and integrating only the effective elements from both frameworks. In

---

**Algorithm 2** Pipeline of MaxEnt-Veritas

---

**Require**: Initialize the RL agent as $\pi_0$.

1: **for** $t = 0, 1 \cdots T - 1$ **do**
2:    Samples $\{\pi_0, \pi_1 \cdots \pi_t\}$ uniformly to induce states.
3:    Define intrinsic reward functions $r_t(s) = \log(\|s - s^{kNN}\|_2^p)$ based on states induced by $\{\pi_0, \pi_1 \cdots \pi_t\}$, same to $r_t^{LA}(s)$.
4:    Trains a RL agent to get $\pi_{t+1}$ which can maximize $r_t(s)$ .
5: **end for**
6: **return** $\{\pi_0, \pi_1 \cdots \pi_T\}$.

---

essence, the procedure involves iteratively updating $r_t^{LA}(s)$ and then training the RL agent with this reward until convergence, as outlined in Algorithm 2. For Step 4 in Algorithm 2, we initialize the RL agent with the policy that achieves the highest score, as evaluated by the current reward function, from the set $\{\pi_0, \pi_1, \ldots, \pi_t\}$. .

Intuitively, this algorithm encourages agents to avoid previously visited states while updating the long-term memory of "visited states" more gradually, resembling Baars' global workspace theory (Baars, 1988). Theoretically, Algorithm 2 is a provably efficient method for state entropy maximization:

**Theorem 1** *We assume that the reward functional $R(d_\pi(s)) = H_{kNN}(d_\pi(s))$ is $\beta$-smooth and $B$-bounded, where $d_\pi(s)$ is approximated with kNN density estimator. Additionally, we assume that the RL agent at the iteration $t$ in Algorithm 2 converges to a $\varepsilon_{1,t}$-optimal solution with locally stationary reward functions $r_t(s) = \log(\|s - s^{kNN}\|_2)$, and the estimation error of the state distribution is $\varepsilon_{0,t}$. When Algorithm 2 is run for $T$ iterations , we have that:*

$$R(d_{\pi_{mix},T+1}) \geq \max_\pi R(d_\pi) - \varepsilon \tag{11}$$

*in which*

$$\varepsilon = \frac{B}{T+2} + 2\beta\bar{\varepsilon}_0 + \bar{\varepsilon}_1 + \frac{\beta}{T+2}[\rho + ln(T+2) + \epsilon_{T+2}] \tag{12}$$

*where $\bar{\varepsilon}_0 = \frac{(\varepsilon_{0,0} + \varepsilon_{0,1} \cdots + \varepsilon_{0,T})}{T+2}$ is the average estimation error of state distribution, $\bar{\varepsilon}_1 = \frac{(\varepsilon_{1,0} + \varepsilon_{1,1} \cdots + \varepsilon_{1,T})}{T+2}$ is the average training error given reward functions over all iterations, $\rho < 0.58$ is the Euler-Mascheroni constant and $\epsilon_T \leq \frac{1}{8T^2}$ which approaches 0 as $T$ goes to infinity. ,*

The value of $\varepsilon$ is determined by $T$, $\bar{\varepsilon}_1$, and $\bar{\varepsilon}_0$. The gap can be reduced by either increasing the number of iterations or minimizing the training/estimation error in each iteration, which aligns well with intuition. If we further assume access to $\varepsilon_0$-optimal estimation oracles and $\varepsilon_1$-optimal planning oracles, as in MaxEnt-H (Lemma 1), $\bar{\varepsilon}_0$ and $\bar{\varepsilon}_1$ are constants. Under these conditions, $\varepsilon$ decreases approximately at a rate of $\frac{B + \beta \ln T}{T}$ as $T$ increases.

It is important to note that Theorem 1 differs significantly from the main theorem of MaxEnt-H, which can be expressed as $\varepsilon = Be^{-T\eta} + 2\beta\varepsilon_0 + \varepsilon_1 + \eta\beta$ (Hazan et al., 2019). When $\eta \to 0$ in MaxEnt-H, the policy selection method can be thought of as uniform sampling. In this scenario, however, $\varepsilon$ does not decrease as $T$ increases. This self-contradiction arises from the multiplication by $\frac{1}{\eta}$ during the derivation of $\varepsilon$.

Furthermore, Theorem 1 provides a clear explanation as to why we assert that one-step updates, as in MaxEnt-LA, are not optimal. If only one step is updated in each iteration, with a different $r_t(s)$, it becomes exceedingly difficult to guarantee an acceptable tolerance $\varepsilon_1$. Of course, when $r_t(s)$ changes much more slowly than the convergence speed of the DRL agent, it is feasible to train the RL agent for only a small number of steps in each iteration. This explains why MaxEnt-LA performs well in many scenarios. However, this trade-off between non-stationarity and the number of training steps per iteration must be carefully fine-tuned based on the specific application.

## 5 EMPIRICAL ANALYSIS

The experimental section is organized as follow. In **Section 5.1**, we quantitatively illustrate that the sampling method of MaxEnt-H is redundant by conducting ablation studies on $\eta$. Afterwards, we quantitatively demonstrate that MaxEn-V outperforms other MaxEnt frameworks in maximizing state entropy (**Section 5.2**) and state coverage. We implement the Soft Actor Critic (SAC) (Haarnoja et al., 2018b) with never-give-up regularizer (Badia et al., 2020) for Mujoco environments as the respective oracle algorithms. Fig. 1 illustrates the environments in which we conduct our experiments. In Walker2D, the agent can only move forward or backward within a 2D spatial plane, whereas the Ant and Humanoid agent can navigate freely in all directions within a 3D space. Please see **Appendix A.2** for more details on experimental settings.

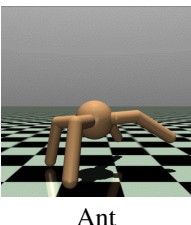 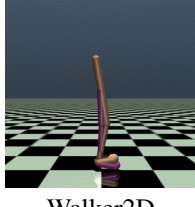 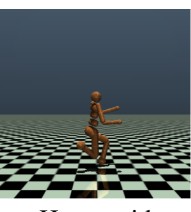

| Ant | Walker2D | Humanoid |

**Figure 1:** Visual interfaces of Mujoco robotic simulations

### 5.1 IS THE $\eta$-BASED SAMPLING METHOD REDUNDANT IN NON-TABULAR ENVIRONMENTS?

Recall that Theorem 2 demonstrates that a very small $\eta$ must be selected to achieve any meaningful tolerance with the theoretical guarantees provided by MaxEnt-H. With such a small value of $\eta$, the sampling method proposed in MaxEnt-H essentially behaves as uniform sampling when $T$ is not significantly large. Thus, we contend that $\eta$-based sampling is redundant. In this section, we empirically validate this assertion. As discussed in Section 3, the sum of probabilities based on $\eta$ is not equal to 1. Consequently, the official MaxEnt-H implementation utilizes the CVXPY Python package (Diamond & Boyd, 2016) to provide an approximate solution by solving the following optimization problem:

$$\arg\min_{\mathbf{x}_t}(\mathbf{x}_t - [\alpha_0, \alpha_1 \cdots \alpha_t])$$
$$\text{subject to } \mathbf{x}_t > \mathbf{0}, \; |\mathbf{x}_t| = 1 \tag{13}$$

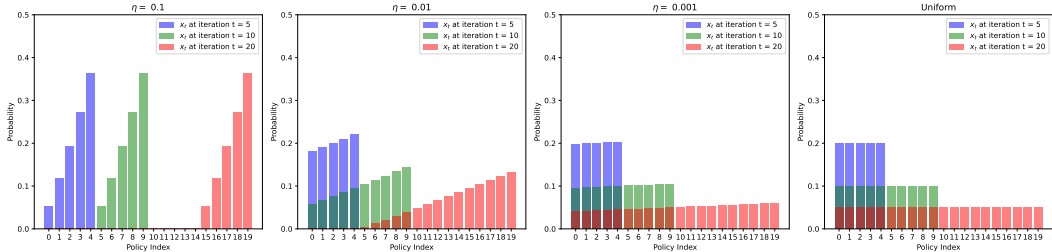

**Figure 2:** The practical sampling probabilities of MaxEnt-H given different $\eta$ at iteration 5, 10, and 20. Smaller the $\eta$, closer to the uniform sampling. If we wants a meaningful guarentee where $\eta < 0.00136$, the sampling method is already very close to uniform sampling.

MaxEnt-H then uses the values within $\mathbf{x}_t$ as the sampling probabilities at iteration $t$. We present the practical sampling probabilities $\mathbf{x}_t$ in iterations $t = 5, 10, 20$ for $\eta = [0.1, 0.01, 0.001]$ in Fig. 2. As illustrated in Fig. 2, when the official MaxEnt-H implementation selects $\eta = 0.1$, the sampling method based on $\eta$ predominantly samples recent policies. However, to guarantee any meaningful tolerance (Theorem 2), the method effectively resorts to uniform sampling (see $\eta = 0.001$ in Fig. 2).

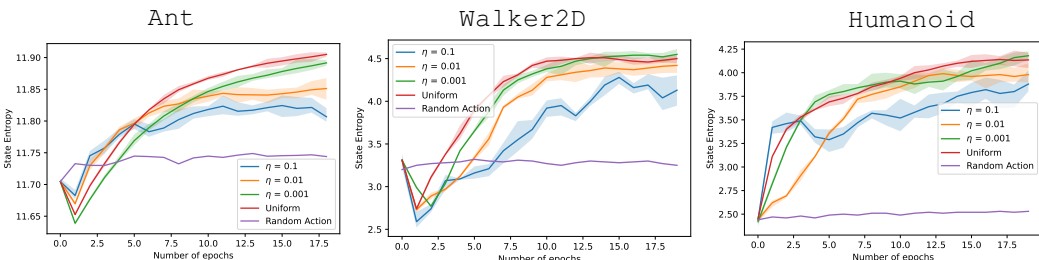

**Figure 3:** Results of different $\eta$. The Y-axis shows the state entropy of the policy evolving with the number of epochs. The experimental settings are identical to the official MaxEnt-H implementation.

More critically, we perform an ablation study on $\eta$ using the official MaxEnt-H implementation[1]. We examine $\eta = [0.1, 0.01, 0.001]$ alongside uniform sampling, where $\eta = 0.1$ corresponds to the value utilized in the official MaxEnt-H implementation, and $\eta = 0.001$ guarantees a meaningful tolerance according to Theorem 2. As illustrated in Fig. 3, uniform sampling consistently surpasses the other $\eta$ values in terms of both entropy and the monotonicity of the learning curves. Consequently, we empirically demonstrate that the $\eta$-based sampling method is redundant.

## 5.2 RESULTS OF STATE ENTROPY AND STATE COVERAGE

In the following, we compare our approach with the two other MaxEnt frameworks, using the number of unique visited states and the state entropy induced by all policies throughout the entire training process as evaluation metrics. Given the continuous high-dimensional state spaces, counting visited states becomes practically challenging. To address this during probability estimation, we reduced the state vectors to a 7-dimensional representation by combining the agent's location $x$-$y$ or $x$-$z$ in the grid with a 5-dimensional random projection of the remaining variables. The distribution $d_\pi(s)$ is estimated using the $k$NN density estimation, with $k$ fixed at 3. For illustration, all methods are evaluated using the same histogram structure by selecting the $x$-$y$ or $x$-$z$ coordinates, bounded by $[[-40, 40], [-40, 40]]$ for Ant, $[[-20, 5], [0.5, 2]]$ for Walker2D, and $[[-10, 10], [-10, 10]]$ for Humanoid. Within these bounds, we assign all samples to a $100 \times 100$ histogram and count the visited states.

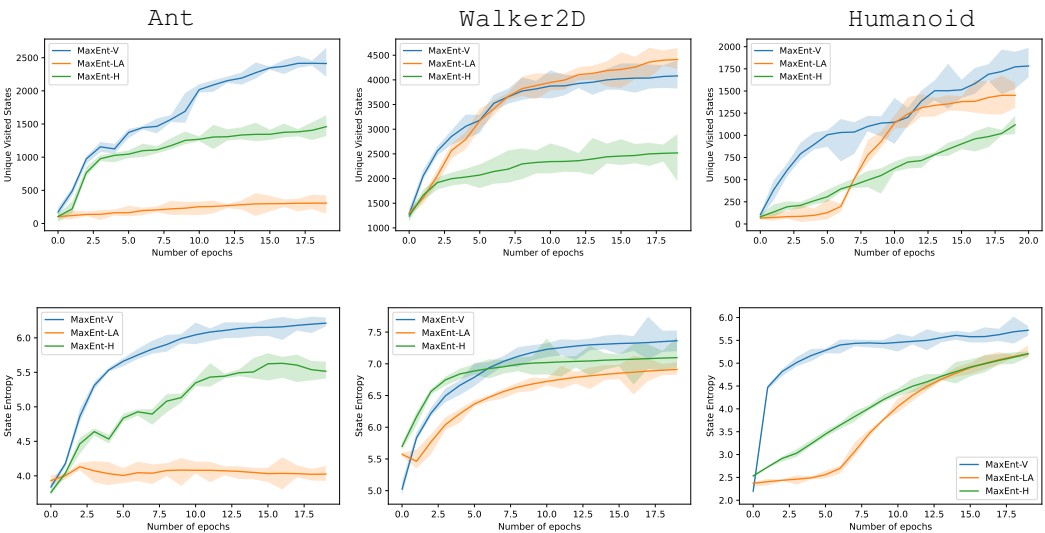

**Figure 4:** Performance of state entropy maximization and state coverage. Evaluated by discrete Shannon entropy and total unique visited states in the training process.

---

[1]https://github.com/abbyvansoest/maxent/tree/master

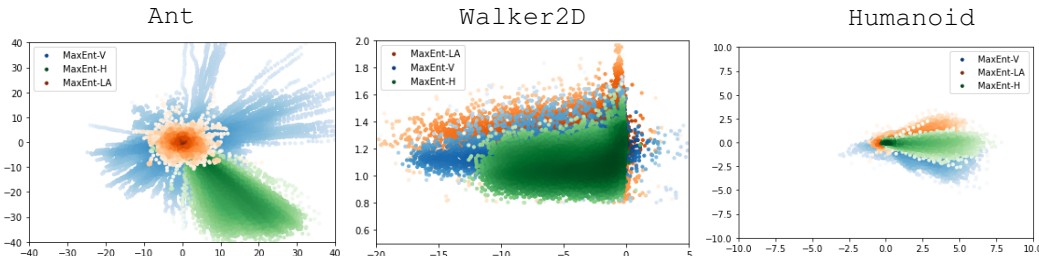

**Figure 5:** The log-probability of occupancy of the two-dimensional state space, corresponding to the maximum entropy achieved by different methods

It is important to note that we do not follow the experimental setup of MaxEnt-H in this section, as it trains the oracles with health constraints but disregards these constraints when using the mixed policy to induce states. This discrepancy leads to training and testing in different environments and allows illegal actions, which may result in unrealistic behaviors, such as the robot being "launched into the sky." To address this, we maintain the health constraints consistently in both training and testing scenarios. Although this approach reduces state coverage, the skills learned are more plausible. **We include videos in the supplementary materials to demonstrate how the agents behave after or during training.** We set $\eta = 0.1$ for MaxEnt-H, consistent with the official MaxEnt-H implementation. The learning curves, with the $y$-axis representing the number of unique visited spatial coordinates and Shannon state entropy values, are shown in Fig. 4. Overall, our method outperforms the baseline approaches in terms of both exploration range and sample efficiency.

Fig. 5 displays the log-probability of occupancy in the two-dimensional state space, corresponding to the maximum entropy achieved by the different methods. The visualization for Ant serves as a clear illustrative example for the three approaches. In the first iteration, the states used for intrinsic reward computation are induced by a random policy $\pi_0$, which are concentrated near the starting point. Since we adopt a $k$NN estimator, the optimal policy, $\pi_1$, in iteration 1 is simply to move as far away from the starting point as possible, i.e., moving in one direction until the time limit is reached. Subsequently, $\pi_2$ is encouraged to stay away from both the starting point and the direction occupied by $\pi_1$. As a result, in each iteration, the optimal policy consistently moves radially in a different direction. If training continues, the MaxEnt-V agent explores the state space in a radial pattern, resembling a "fireworks" effect.

In contrast, MaxEnt-H with $\eta = 0.1$ samples states using only recent policies to define $r_t^H(s)$, quickly forgetting visited states, as shown in Fig. 2. Consequently, its exploration traces are confined within a smaller range. For MaxEnt-LA, the issue arises from updating the intrinsic reward function too frequently, causing the agent to be discouraged from revisiting previously visited states. This method faces a fundamental limitation due to the rapid decay of rewards: once a state is visited, its reward diminishes significantly, preventing the agent from revisiting it, even if it might lead to unexplored downstream states (Bellemare et al., 2016; Stanton & Clune, 2018; Ecoffet et al., 2019; Badia et al., 2020).

In the other two environments, although MaxEnt-V does not dramatically outperform the others in terms of spatial coverage, it learns distinct action modes compared to the other two methods. In Walker2D, MaxEnt-V is the only approach that learns to move forward, as indicated by the points with $x$ values greater than 0 in Fig. 5. In Humanoid, MaxEnt-V is the only method that learns to move backward, represented by points with $x < 0$ in Fig. 5.

## 6    RELATED WORKS

Before introducing reinforcement learning (RL) methods for exploration, it is essential to clarify the distinction between maximum state entropy and the well-known Soft Q-learning and Soft Actor-Critic (SAC) algorithms (Haarnoja et al., 2017; 2018a). These methods hypothesize that robust policies can be learned by exploring the policy space and provide a framework for maximizing both extrinsic rewards and policy entropy $H(\pi(\mathbf{a}|\mathbf{s}))$, with theoretically grounded policy improvement guarantees. However, these methods (Haarnoja et al., 2018b; Yang et al., 2021; Eysenbach & Levine,

2021) lack the capability to explore environments in the absence of extrinsic rewards. To address this limitation, Hazan et al. (2019) has suggested that exploration agents should instead maximize a convex entropy function of the visitation distribution over the state space, i.e., $H(\mathbf{s})$.

**State Entropy Maximization for Exploration** Following MaxEnt-H, several variants have been proposed in recent years. Its Rényi variant (Zhang et al., 2021) follows a similar structure, with only minor adjustment on the reward function, i.e., $\hat{r}(\mathbf{s}) = \log \hat{p}^{(\alpha-1)}(\mathbf{s})$. Other improvements includes integration of representation learning (Nedergaard & Cook, 2022; Yarats et al., 2021), and efforts to reduce sample complexity (Tiapkin et al., 2023), just to name a few. In contrast, MaxEnt-LA can be considered as an intrinsic learning method. In scenarios where extrinsic rewards are unavailable intrinsic exploration aims to develop an intrinsic reward function as a substitute. This make these methods seamlessly embrace any existing RL algorithms by simply changing rewards. For non-tabular state entropy maximization without learning probability density models, RE3 (Seo et al., 2021) propose to implement random encoder instead of a pre-trained one via contrastive learning. After that, RISE (Yuan et al., 2023) extends it to Rényi entropy. Recently, Kim proposed to maximize the value-conditional state entropy, which separately estimates the state entropies that are conditioned on the value estimates of each state, then maximizes their average (Kim et al., 2024).

Another type of related approaches to "maximum state entropy" (Mutti et al., 2021; Jain et al., 2024) focuses on maximizing trajectory-wise state entropy, which intuitively encourages visiting diverse states within a finite number of steps or within a single episode. Although these methods share a similar name with state entropy maximization, their objectives are fundamentally different. Therefore, in this work, we do not delve deeply into them.

**Parametric Methods for Exploration** In addition to MaxEnt-based non-parametric exploration methods, deep neural network-based parametric methods (Pathak et al., 2017; Ecoffet et al., 2019; Burda et al., 2019; Badia et al., 2020; Dewan et al., 2024) have garnered significant attention in recent years. These methods encourage agents to explore novel states in a non-stationary manner by assigning greater rewards to states that are less frequently visited by estimating predictive forward models and use the prediction error as the intrinsic motivation. These *curiosity*-driven approaches have their roots traced back to the 1970's when Pfaffelhuber introduced the concept of "observer's information" (Pfaffelhuber, 1972) and Lenat (Lenat, 1976) introduced the concept of "interesting-ness" in mathematics to promote the novel hypotheses and concepts (Amin et al., 2021). Recently, popular prediction error-based approaches fall under this category. The recent surge in popularity of these networks is strongly related to advancements in deep neural networks (DNNs). For instance, ICM (Pathak et al., 2017) and RND (Burda et al., 2019), utilize a CNN as the internal model to predict the next image, while GIRIL implements a variational autoencoder (VAE) to model transitions in environments. After that, some approaches find the novelty vanishing problem and try to solve it by introducing an episodic mechanism (Ecoffet et al., 2019; Badia et al., 2020).

## 7 CONCLUSION

In this paper, we analyze two fundamental approaches for state entropy maximization in reinforcement learning. We find that the $\eta$-based sampling, a key procedure in MaxEnt-H, is superfluous for achieving any meaningful tolerance. In contrast, MaxEnt-LA updates its intrinsic reward function too frequently, resulting in the agent being encouraged to maximize different reward functions at each step, which makes it difficult to explore a broader state space.

This rethinking leads to a simple method that incorporates only the efficient components of both approaches, which we term MaxEnt-(V)eritas. Compared to MaxEnt-H, it mainly replaces the $\eta$-based sampling with uniform sampling. For MaxEnt-LA, MaxEnt-V updates the reward function gradually rather than at every training step.

We empirically validate our analysis and evaluate MaxEnt-V in three robotic Mujoco environments. An ablation study on $\eta$ demonstrates that better results are achieved as $\eta \to 0$, corresponding to uniform sampling. Additionally, MaxEnt-V significantly outperforms the baseline methods in terms of state coverage and state entropy maximization.

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

# A APPENDIX

## A.1 PROOFS

### A.1.1 PROOFS OF PROPOSITION 2

*Proof.*

Given Lemma 1, we have $\eta = \frac{0.1\varepsilon^2}{40|S|}$. To obtain any $\varepsilon < \log|\mathcal{S}|$, we have:

$$\eta < \frac{\log^2(|\mathcal{S}|)}{400|\mathcal{S}|} \tag{14}$$

For any $|S| \geq 2$, we find the maximum value of $\frac{\log^2(|\mathcal{S}|)}{400|\mathcal{S}|}$ in the following. Let $x = |S|$, we first find the first and second derivative of the function $f(x) = \frac{\log^2(x)}{400x}$:

$$f'(x) = \frac{2\log(x) - (log(x))^2}{400x^2} \tag{15}$$

$$f''(x) = \frac{3\log(x) - 1}{200x^3} \tag{16}$$

The function is convex when:

$$f''(x) = \frac{3\log(x) - 1}{200x^3} \geq 0 \tag{17}$$

That is:

$$x \geq e^{1/3} \approx 1.37 \tag{18}$$

Thus, for any $x = |S| \geq 2$, the function is convex. Let $f'(x) = 0$, we have the maximum value of the function to be:

$$f(e^2) = \frac{\log^2(e^2)}{400e^2} < 0.00136 \tag{19}$$

So,

$$\eta < \frac{\log^2(|\mathcal{S}|)}{400|\mathcal{S}|} < \max[\frac{\log^2(|\mathcal{S}|)}{400|\mathcal{S}|}] < 0.00136 \tag{20}$$

### A.1.2 PROOFS OF PROPOSITION 3

*Proof.*

Given Lemma 1, we have $T > \frac{40|S|}{0.1\varepsilon^2}\log\frac{\log|S|}{0.1\varepsilon}$ to guarantee the tolerance $\varepsilon$. To obtain any $\varepsilon < \log|\mathcal{S}|$, we have:

$$T > \frac{40|S|}{0.1\log^2|\mathcal{S}|}\log\frac{\log|S|}{0.1\log|\mathcal{S}|}$$
$$T > \frac{1329|S|}{\log^2|S|} \tag{21}$$

### A.1.3 PROOFS OF THEOREM 1

*Proof.*

We assume that the reward functional $R = H_{kNN}$ is $\beta$-smooth, $B$-bounded, for all $X, Y$.

$$\|\nabla R(X) - \nabla R(Y)\|_\infty \leq \beta\|X - Y\|_\infty \tag{22}$$

$$-\beta\mathbb{I} \preceq \nabla^2 R(X) \preceq \beta\mathbb{I}; \quad \|\nabla R(X)\|_\infty \leq B \tag{23}$$

Let $\pi^*$ be the optimal policy, we have (Hazan et al., 2019):

$$R(d_{\pi_{\mathrm{mix},t+1}}) = R((1 - \frac{1}{t+2})d_{\pi_{\mathrm{mix},t}} + \frac{1}{t+2}d_{\pi_{t+1}}) \qquad \text{Equation 3}$$

$$\geq R(d_{\pi_{\mathrm{mix},t}}) + \frac{1}{t+2}\langle d_{\pi_{t+1}} - d_{\pi_{\mathrm{mix},t}}, \nabla R(d_{\pi_{\mathrm{mix},t}})\rangle - (\frac{1}{t+2})^2\beta\|d_{\pi_{t+1}} - d_{\pi_{\mathrm{mix},t}}\|_2^2 \quad \text{smoothness}$$

where

$$\langle d_{\pi_{t+1}}, \nabla R(d_{\pi_{\text{mix},t}})\rangle \geq \langle d_{\pi_{t+1}}, \nabla R(\hat{d}_{\pi_{\text{mix},t}})\rangle - \beta\|d_{\pi_{\text{mix},t}} - \hat{d}_{\pi_{\text{mix},t}}\|_\infty$$

$$\geq \langle d_{\pi^*}, \nabla R(\hat{d}_{\pi_{\text{mix},t}})\rangle - \beta\varepsilon_{0,t} - \varepsilon_{1,t} \geq \langle d_{\pi^*}, \nabla R(d_{\pi_{\text{mix},t}})\rangle - 2\beta\varepsilon_{0,t} - \varepsilon_{1,t}$$

The first and last inequalities is from Eq. (22) (Bubeck et al., 2015), while the second inequality above is given by the conclusion of Theorem 1 which is $\nabla R(\hat{d}_{\pi_{\text{mix},t}}) = r_t^H \propto r_t^{LA}$ and the definition of training error $\varepsilon_{1,t}$. For the optimal policy $\pi^*$:

$$V_{\pi_{t+1}} = \langle d_{\pi_{t+1}}, r_t^{LA}\rangle \geq V_{\pi^*} - \varepsilon_{1,t} = \langle d_{\pi^*}, r_t^{LA}\rangle - \varepsilon_{1,t} \tag{24}$$

Reconsider $R(d_{\pi_{\text{mix},t+1}})$, we have:

$$R(d_{\pi_{\text{mix},t+1}}) \geq R(d_{\pi_{\text{mix},t}}) + \frac{1}{t+2}\langle d_{\pi^*} - d_{\pi_{\text{mix},t}}, \nabla R(d_{\pi_{\text{mix},t}})\rangle - \frac{2}{t+2}\beta\varepsilon_{0,t} - \frac{1}{t+2}\varepsilon_{1,t} - (\frac{1}{t+2})^2\beta$$

$$\geq (1 - \frac{1}{t+2})R(d_{\pi_{\text{mix},t}}) + \frac{1}{t+2}R(d_{\pi^*}) - \frac{2}{t+2}\beta\varepsilon_{0,t} - \frac{1}{t+2}\varepsilon_{1,t} - (\frac{1}{t+2})^2\beta$$

Then,

$$R(d_{\pi^*}) - R(d_{\pi_{\text{mix},t+1}}) \leq (1 - \frac{1}{t+2})(R(d_{\pi^*}) - R(d_{\pi_{\text{mix},t}})) + \frac{2}{t+2}\beta\varepsilon_{0,T} + \frac{1}{t+2}\varepsilon_{1,T} + (\frac{1}{t+2})^2\beta.$$

Thus far, the steps are largely analogous to those in MaxEnt-H. The key differences lie in the definition of $r_t$ and the sampling strategy: we sample each policy with probabilities $\alpha_0 = \alpha_1 = ... = \alpha_{t+1} = 1/(t+2)$, whereas MaxEnt-H defines the probabilities as $\alpha_t = \eta^t$. This distinction leads to markedly different conclusions when telescoping the inequality above:

$$R(d_{\pi^*}) - R(d_{\pi_{\text{mix},T+1}}) \leq (1 - \frac{1}{T+2})(R(d_{\pi^*}) - R(d_{\pi_{\text{mix},T}}))$$

$$+ \frac{2}{T+2}\beta\varepsilon_{0,T} + \frac{1}{T+2}\varepsilon_{1,T} + (\frac{1}{T+2})^2\beta.$$

$$\leq \frac{T+1}{T+2}[\frac{T}{T+1}(R(d_{\pi^*}) - R(d_{\pi_{\text{mix},T-1}}))$$

$$+ \frac{2}{T+1}\beta\varepsilon_{0,T-1} + \frac{1}{T+1}\varepsilon_{1,T-1} + (\frac{1}{T+1})^2\beta]$$

$$+ \frac{2}{T+2}\beta\varepsilon_{0,T} + \frac{1}{T+2}\varepsilon_{1,T} + (\frac{1}{T+2})^2\beta.$$

$$\cdots$$

$$= (\frac{T+1}{T+2} \times \frac{T}{T+1} \cdots \times \frac{1}{2})(R(d_{\pi^*}) - R(d_{\pi_{\text{mix},0}}))$$

$$+ \frac{2\beta}{T+2}\sum_{t=0}^{T}\varepsilon_{0,t} + \frac{1}{T+2}\sum_{t=0}^{T}\varepsilon_{1,t}$$

$$+ \frac{\beta}{T+2}\left[\frac{1}{T+2} + \frac{1}{T+1} \cdots + \frac{1}{2} + 1\right].$$

The last term is a harmonic series, so we have:

$$R(d_{\pi^*}) - R(d_{\pi_{\text{mix},T+1}}) \leq \frac{B}{T+2} + \frac{2\beta\sum_{t=0}^{T}\varepsilon_{0,t}}{T+2} + \frac{\sum_{t=0}^{T}\varepsilon_{1,t}}{T+2}$$

$$+ \frac{\beta}{T+2}[\rho + \ln(T+2) + \epsilon_{T+2}]$$

where $\rho < 0.58$ is the Euler-Mascheroni constant and and $\epsilon_T \leq \frac{1}{8T^2}$ which approaches 0 as $T$ goes to infinity.

## A.2 Details of Experimental Setting

All experiments are conducted on single V-100 GPUs, where the maximum memory usage is up to 5G for each single training process. In Walker2D, the agent can only move forward or backward within a 2D spatial plane, whereas the Ant and Humanoid agent can navigate freely in all directions within a 3D space. Both agents are reset to starting points near $(0, 0)$ if they fail to meet the health conditions specified by the default setting (Brockman et al., 2016). The default number of steps for truncation id fixed as default setting 1000, without any fine-tuning. The details of the three environments are given below.

**Ant** is a three-dimensional robot composed of a single torso, which is a freely rotating body, and four legs connected to it. Each leg consists of two links. The observation is a 29D vector. The 29-dimensional state space was first reduced to dimension 7, combining the agent's $x$ and $y$ location in the gridspace with a 5-dimensional random projection of the remaining 27 states.

**Walker2D** The walker2D is a two-dimensional two-legged figure that consist of seven main body parts - a single torso at the top (with the two legs splitting after the torso), two thighs in the middle below the torso, two legs in the bottom below the thighs, and two feet attached to the legs on which the entire body rests. The observation is a 18D vector. The 18-dimensional state space was first reduced to dimension 7, combining the agent's $x$ and $z$ location in the gridspace with a 5-dimensional random projection of the remaining 16 states.

**Humanoid** The 3D bipedal robot is designed to simulate a human. It has a torso (abdomen) with a pair of legs and arms. The legs each consist of three body parts, and the arms 2 body parts (representing the knees and elbows respectively). The observation is a 378D vector. The 378-dimensional state space was first reduced to dimension 7, combining the agent's $x$ and $y$ location in the gridspace with a 5-dimensional random projection of the remaining 376 states.

The random encoders implemented in this work have been widely adopted by previous methods (Seo et al., 2021; Hazan et al., 2019; Kim et al., 2024).

| Hyper-parameters | Value |
|---|---|
| initial temperature | 0.2 |
| gamma | 0.99 |
| actor_lr | 3e-4 |
| critic_lr | 3e-4 |
| q_lr | 3e-4 |
| soft_update_rate | 0.005 |
| hidden_dim | 256 |
| memory size | 1e+6 |
| layer_num | 3 |
| batch_size | 128 |
| layer_num | 3 |
| activation_function | torch.relu |
| last_activation | None |

**Table 1:** Hyper-parameters of SAC

We adopt SAC as backbones. For the SAC[2] oracle, we summarizes our hyper-parameters in Table 1.

In this paper, particularly in the experiment section, we choose Soft Actor-Critic (SAC) as our oracles. Here, we will succinctly outline the key equations of SAC. Diverging from the standard MDP, SAC incorporates a policy entropy term to enhance exploration within the conditioned action space, i.e., $\max[r + \beta H(\pi(a|s))]$ where $\beta$ is temperature.It is crucial to note that the policy entropy term used in SAC is distinct from the state entropy concept discussed in our study. This distinction arises from the different domains in which these two entropy forms operate. While the policy entropy in SAC focuses on the conditional action-selection process, the state entropy we examine pertains to the diversity of state visitations. This clarification is essential for understanding the unique con-

---

[2]https://github.com/seolhokim/Mujoco-Pytorch

tributions and applications of each entropy type within the realm of reinforcement learning. SAC iteratively update critic using soft $Q$ function $Q_\phi$ and actor $\pi_\theta$ by minimizing the KL divergence between the soft value function and policy distribution. Besides, temperature is also adaptive. More formally, three objective functions are:

$$
\begin{aligned}
J_Q(Q_\phi, r) = \mathbb{E}_{\{s_{t+1}, s_t, a_t, r_t\} \sim D}[(Q_\phi(s_t, a_t) - r_t \\
- \gamma(Q_{\hat{\phi}}(s_{t+1}, \pi_\theta(s_{t+1})) - \beta log\pi_\theta(s_{t+1})))^2]
\end{aligned}
\tag{25}
$$

$$
J_\pi(\pi_\theta) = \mathbb{E}_{s_t \sim D}[-\gamma(Q_\phi(s_t, \pi_\theta(s_t)) - \beta log\pi_\theta(s_t))]
\tag{26}
$$

$$
J(\beta) = \mathbb{E}_{s_t \sim D}[-\beta(\hat{H} + log\pi_\theta(s_t))]
\tag{27}
$$

where $D$ denotes replay buffer, $\hat{H}$ is the expected target policy entropy, $Q_{\hat{\phi}}$ is the target critic deep neural network.

In practice, we found that the Never-Give-Up regularizer (Badia et al., 2020) is highly effective in preventing SAC from converging to local optima. Specifically, a $k$NN-based term, $\sum_{s^{kNN} \in \tau} \|s - s^{k\text{NN}}\|_2$, is introduced to the reward functions of MaxEnt-H, MaxEnt-LA, and MaxEnt-V.

