# OpenReview forum: "RETHINK MAXIMUM STATE ENTROPY"
_ICLR.cc/2025/Conference — Submitted to ICLR 2025_

### Official Review · Reviewer_YseA · 2024-10-24

**Soundness:** 1
**Presentation:** 3
**Contribution:** 2
**Rating:** 3
**Confidence:** 4

**Summary:**

This paper addresses maximum state entropy exploration in MDPs without rewards. First, it analyzes and compare previous approaches to state entropy maximization, namely a Frank-Wolfe algorithm (like MaxEnt, Hazan et al. 2019) and a policy optimization algorithm (like APT, Liu & Abbeel, 2021). The paper shows that the two approaches share the same reward function when the entropy is estimated via kNN. Then, the paper proposes a new algorithm, called MaxEnt-V, incorporating the best of both approaches. The method is demonstrated to lead to approximately optimal policies and empirically validated in a set of Mujoco environments.

**Strengths:**

- The paper provides a unifying perspective on the literature of maximum state entropy;
- The paper proposes an interesting implementation change for maximum state entropy algorithms, which prescribes to freeze the intrinsic reward for multiple policy optimization steps (like in a Frank-Wolfe routine);
- The paper makes an effort to provide theoretical ground to the proposed method.

**Weaknesses:**

EVALUATION

This paper provides some fresh ideas, mostly that the previous approaches for maximum state entropy can be connected into a "unifying" algorithm and that the entropy intrinsic reward shall be frozen for multiple policy optimization steps, which bears some resemblance to the target network trick for deep Q-learning. Unfortunately, in my opinion the paper fails to provide convincing evidence that the frozen reward trick gives empirical benefit to justify widespread adoption. More broadly, I think the paper falls short of the technical quality required to be accepted at ICLR. I provide below a summary of what I believe are the main weaknesses of the paper and further comments.

MAJOR WEAKNESSES

1) Misunderstandings of the literature
   1. The paper presents MaxEnt-H and MaxEnt-LA as two competing approaches for maximum state entropy. Whereas different design choices can be extracted on an abstract level, the authors seem to misunderstand the purpose of the two papers. Hazan et al. were the first to introduce the problem of maximum state entropy in MDPs. They show the latter problem is not hopeless, despite being non-convex in the policy parameters, by providing a provably efficient algorithm. While an implementation and experiments are provided, the algorithm is mostly a theoretical and analytical tool. Instead, Liu & Abbeel's paper falls into a stream of practical methods for maximum state entropy, where the main advancements were given by the use of kNN entropy estimators in place of state densities (this technique has been introduced by Mutti et al., 2021 not by Liu & Abbeel as stated in the manuscript) and learning representations.
   2. The paper suggests that MaxEnt-LA trains a mixture with uniform parameters. I am not sure this is the case. From my understanding, it does sample uniformly from a replay buffer, which may include transitions coming from previous policies, but it does that only to perform updates on the *last* policy. Indeed, the output is the last policy network, which is then fine-tuned with external reward, and not a mixture. There seems to be an important gap here.
   3. The paper says that another stream of work (Mutti et al 2021, Jain et al 2024) optimize the trajectory-wise entropy. Perhaps this distinction shall be clarified. From my understanding, the objective of Mutti et al 2021 is the same of Eq. 2, although the entropy reward is not decomposed in state terms as in Liu & Abbeel 2021.

2) Some claims look subjective and lack strong support
   1. "Tuning $\eta$ is unnecessary in MaxEnt-H". Since the purpose of tuning $\eta$ comes from the analysis in Hazan et al., 2019, the paper should show that similar convergence guarantees  can be obtained with a uniform mixture to support this claim. Showing that $\eta$ is always small and tuning is unnecessary in practice is not enough.
   2. "Freezing the reward is better". Theorem 1 only provides an upper bound on the sub-optimality. It is rather weak to say that freezing the reward is better because it leads to a smaller upper bound. Maybe the upper bounds are just not tight, and the analysis would say nothing about which one (freezing the reward or changing it at any step) is better.
   3. Theorem 1. I have some concerns on the validity of this result. First, the statement assumes that $H_{kNN}$ is smooth and bounded, which does not seem to be the case by staring at Eq. 6. Can the authors provide more details on when those conditions are met? Moreover, what does it mean to assume access to estimation oracle (as in lien 310)? Note that the kNN entropy estimators are biased, does that mean $\epsilon_0$ error on the biased estimate or the true entropy?

3) The empirical analysis looks very far from the standards of the community
   - Are the curves in the figures reported on a single run? Some details seem to be missing on how the experiments are conducted to meet some statistical significance (e.g., look at https://ojs.aaai.org/index.php/AAAI/article/view/11694 and https://proceedings.neurips.cc/paper_files/paper/2021/hash/f514cec81cb148559cf475e7426eed5e-Abstract.html)
   - Previous papers, especially Hazan et al. 2019 and Liu & Abbeel 2021 have public implementations. To claim MaxEnt-V gives benefits over them, it would be better to compare its performance with the official implementations of MaxEnt-H and MaxEnt-LA.

OTHER COMMENTS

The literature of maximum state entropy could be presented better, especially considering that this manuscript builds over them. The first papers on this problem have been Hazan et al 2019; Lee et al 2019, Mutti & Restelli 2020), which presented algorithms requiring state density estimation, which is mostly impractical in high dimensions. To overcome this issue, Mutti et al 2021 proposed to use kNN entropy estimators (Singh et al 2003 and others) to guide policy optimization without explicit state density estimation. Mutti et al 2021 compute kNN distances on the state features, which is not suited for images. Liu & Abbeel 2021 coupled kNN estimators with contrastive representations (together with various other implementation changes, such as state-based rewards, actor-critic architecture with replay buffers). Other representations have been proposed by subsequent works, such as Seo et al 2021, Yarats et al 2021. Several other works followed on both practical methodologies and theoretical analysis of maximum state entropy. Some relevant references that does not seem to be mentioned:
- Lee et al., Efficient exploration via state marginal matching, 2019;
- Mutti & Restelli, An intrinsically-motivated approach for learning highly exploring and fast mixing policies, 2020;
- Guo et al., Geometric entropic exploration, 2021;
- Liu & Abbeel, Aps: Active pretraining with successor features, 2021;
- Mutti et al., Unsupervised reinforcement learning in multiple environments, 2022;
- Mutti et al., The importance of non-Markovianity in maximum state entropy exploration, 2022;
- Mutti, Unsupervised reinforcement learning via state entropy maximization, 2023;
- Yang & Spaan, CEM: Constrained entropy maximization for task-agnostic safe exploration, 2023;
- Zisselman et al., Explore to generalize in zero-shot rl, 2023;
- Zamboni et al., How to explore with belief: State entropy maximization in pomdps, 2024;
- Zamboni et al., The limits of pure exploration in POMDPs: When the observation entropy is enough, 2024.

Despite my concerns expressed above, I think the idea of the frozen rewards to improve maximum state entropy approaches is very interesting and worth studying. Perhaps the authors could think of restructuring the paper and their analysis to focus on the empirical benefit that this trick may provide and the (stability) issues that may arise from chasing the non-stationary reward.

MINOR
- Perhaps clarify the meaning of MaxEnt-H and MaxEnt-LA earlier in the text
- l.48 "While these importance sampling-based methods" -> what does that mean?
- Eq. 2 min -> max
- Algorithm 2, line 2: how many states are sampled?

**Questions:**

I mostly do not have direct questions. Some are reported in the comments above.

---

> ### Author Response · Authors · 2024-11-19
>
> We appreciate your detailed response; however, we respectfully disagree that the paper is limited. We believe there has been a misinterpretation of our work, as well as of several influential contributions in the MaxEnt literature, due to a focus only on Mutti's work in a non-standard manner, leading to conclusions based on factual errors. We elaborate on these points in detail as follows:
>
> **Misunderstandings of the literature**
>
> 1.	The description "MaxEnt-H is mostly a theoretical tool, while MaxEnt-LA falls into a stream of practical methods" is neither formal nor accurate. One of the main contributions of our paper is to prevent such misunderstandings. We establish a theoretical and algorithmic relationship between these two methods, rather than presenting an abstract distinction, providing a step-by-step explanation. Both methods can transition into one another by adjusting hyperparameters (such as $\eta$, the frequency of updating the replay buffer, and the initialization of $\pi$ in each epoch).
>
> Crucially, MEPOL (Mutti et al., 2021) is **NOT** premilitary to MaxEnt-LA. MaxEnt-H is **NOT** premilitary to MEPOL either. MEPOL dose **NOT** address the same problem.
>
> From a motivational perspective, MEPOL seeks to maximize the entropy of the average state distribution $H\left(\frac{1}{T}\sum_{t=1}^{T}d_t^{\pi}\right)$, defined over episodes (note the index $t$ within an episode), while MaxEnt aims to directly maximize the state distribution $H(d^{\pi})$. Theoretically, MEPOL reduces the general MaxEnt problem to a finite-horizon setting, which requires a fixed hyperparameter (episode length) $T$ for each training process. In experiments, MEPOL has to remove the default termination rules for safe learning in Mujoco to maintain a fixed horizon length, which leads to illegal poses. In contrast, unconstrained MaxEnt methods allow for varying episode lengths in each run. Intuitively, MEPOL partly encourages exploration of novel episodes or trajectories, while MaxEnt focuses on exploring novel states.
>
> From an information-theoretic perspective, MEPOL establishes a projection from the state space to a space defined by the average probabilities within trajectories. Such projections noramlly results in fundamentally different information-theoretic metrics and properties. Further details on these information-theoretic definitions may be found in the follows.
>
> Alfréd Rényi. On measures of entropy and information. 1961.
>
> Principe, Jose C. Information theoretic learning: Renyi's entropy and kernel perspectives. Springer Science & Business Media, 2010.
>
>
>
> From an implementation perspective, MEPOL fundamentally does not present the algorithmic ideas used in Liu et al. or even MaxEnt-H. The key component of MaxEnt-LA is simply a non-stationary intrinsic reward function, which can be integrated into any off-policy RL algorithm (such as Q-learning, DQN, SAC, etc.) by substituting or augmenting the extrinsic rewards, much like the Approximation Oracles in MaxEnt-H. In contrast, the core of MEPOL is an implicit expression of the gradient of the trajectory-wise state entropy defined by itself, constrained by a KL divergence, within the framework of TRPO.
>
> Therefore, it is not correct to position MEPOL between MaxEnt-LA and MaxEnt-H. It is sufficient to mention MEPOL briefly, along with its subsequent works, in the related works section.
>
> 2.  The key component of MaxEnt-LA is a non-stationary intrinsic reward function, which can be implemented into any off-policy RL algorithm. During the training process, it iteratively appends a new sample $(s, s', a, r, d)$ to the replay buffer and updates SAC for one step. Consequently, the mixture of policies corresponds to the SAC models at all checkpoints. The induced states are simply the states visited during the training process. We will also clarify this tip in our paper to prevent confusion among readers who may not be familiar with DRL methods and experimental procedures.
>
> 3. This has been addressed in the first point. The finite horizon setting fundamentally alters the problem, particularly in relatively realistic environments such as Mujoco.

---

> ### Author Response · Authors · 2024-11-19
>
> **Some claims look subjective and lack strong support**
>
> 1.	This is presented in Theorem 1 of our paper. MaxEnt-V can be thought of as MaxEnt-H with uniform sampling. The only differences lie in the definition of the reward function and the method of initializing $\pi$ in each iteration. Besides, another major problem of $\eta$ is that the sampling probability $\alpha_t = \eta(1-\eta)^{t-1}$, with a fixed $\eta$, can never satisfy the condition $\sum_{t=0}^T \alpha_t = 1$. More details and corresponding experiments can be found in line 231 to 239 and line 352 to 377.
> 2.	You have completely misunderstood Theorem 1 (as well as the main theorem of MaxEnt-H), which provides a theoretically grounded gap $\varepsilon$ between the current policy and the **global** theoretical optimal policy, similar to MaxEnt-H, rather than an "upper bound on the sub-optimality." "Freezing the reward" ensures that SACs are trained for an adequate number of steps to optimize subproblems within the Frank-Wolfe-like algorithm, rather than pursuing a smaller upper bound. We recommend a thorough review of Lemma 1 and Theorem 1 for clarification.
> 3.	We do not assume strict smoothness and boundedness, but rather constrained with $B$ and $\beta$. $\beta$-smoothness and $B$-boundedness are not solely determined by Eq. (6) without knowing the distribution of kNN distances. Given the data points, we can always identify the maximum gradient difference $B$ and the maximum smoothness factor $\beta$. The key issue is whether $B$ and $\beta$ are appropriate for the specific environment. In Mickey-Mouse settings, such as in small DAG, data points are sparse, and there are no constraints on distribution shifts between adjacent nodes, leading to ridiculously large of $B$ and $\beta$. In contrast, in robotic environments, actions and states are continuous, with millions of dense data points, and the underlying $B$ and $\beta$ cannot exceed physical constraints such as velocity or acceleration. Thus, it is quite reasonable to assume this in a complex continuous environment.
>
> For $\varepsilon_0$, the kNN entropy estimator consists of the kNN density estimator (see Eq. (9)) and a Monte Carlo estimate of the Shannon entropy ($-1/N \sum \log d$ and $-1 \sum d \log d$). $\varepsilon_0$ represents the error of the density estimates. The expression $H_{kNN} = -1/N \sum \log p$ is not merely an estimate of Shannon entropy, but also represents the particle entropy itself (refer to \cite{ref} for a detailed explanation of what constitutes an 'entropy'). Therefore, Theorem 1 provides tolerance for this entropy without considering the Monte Carlo estimate. We will clarify this by adding a hat to Eq. (6) and the corresponding explanation in the text.
>
>
> **The empirical analysis looks very far from the standards of the community**
>
> It is not true
>
> 1.	The curves we provide represent the average scores from 4 runs. In the revised PDF, we have updated the curves along with the corresponding variances.
>
> 2.	We have implemented the official versions of MaxEnt-H and MaxEnt-LA, using the same evaluation metric and Mujoco setup as in MaxEnt-H. The only difference is that we consistently use the default settings of Mujoco. In contrast to MaxEnt-H's experiments, which train with safety constraints but inducing states without them—an apparent inconsistency—our approach ensures a fair comparison by applying the same conditions throughout. In fact, we are the first to evaluate these two most important MaxEnt methods within the same empirical setting in a consistent manner, eliminating any auxiliary modules such as pre-trained encoders.

---

> ### Author Response · Authors · 2024-11-19
>
> **Other Comments**
>
> We humbly disagree with the way in which you presented the literature, as it does not consider any detailed algorithmic steps or theoretical relationships. In our framework, Hazan et al. (2019) provide the foundational algorithmic and theoretical structure, with the redundant $\eta$. This structure is not limited to entropy but applies to all smooth and bounded reward functions. We show that MaxEnt-LA can be seen as a special case of MaxEnt-H, achieved by using the kNN density estimator, setting $\eta \rightarrow 0$, and updating memory at each step. With the bridge between MaxEnt-H and MaxEnt-LA, other works you mentioned can be easily connected. For example, the difference between Lee et al. (2019) and MaxEnt-LA lies solely in the intrinsic reward function, i.e., KL divergence to a uniform distribution versus kNN entropy. RE3 simply replaces the pre-trained encoder with a random one compared to MaxEnt-LA. Yarats et al. (2021) further improves the encoder for visual representation by iteratively training the encoder and exploring with MaxEnt-LA. Each of these methods can be seen as a modification of the others, all within the framework of Algorithm 1 and Theorem 1. We show that MaxEnt-H and our MaxEnt-V theoretically support MaxEnt-LA and other related works.
>
> In contrast, MEPOL and Guo et al (2021) reduce the problem by imposing a finite-horizon constraint. Mutti and Restelli (2020) constrain the number of steps, and Mutti et al. (The Importance...) assume a unique optimal action within a finite-horizon MDP. Each of these methods shifts the general MaxEnt problem into a different one.
>
> In summary, the reviewer demonstrates a general understanding of MaxEnt methods but lacks a deep understanding of the details, except a few related works by Mutti et al. This has led to several misunderstandings regarding the literature. This highlights the importance of our paper: such misunderstandings would not arise if the reviewer had started from our paper to learn about maximum state entropy, rather than from MaxEnt-H, MaxEnt-LA, or MEPOL, etc.

---

> > ### Comment · Reviewer_YseA · 2024-11-19
> >
> > I thank the authors for the detailed responses. Unfortunately, I found some of my comments to be not fully addressed. I am following up on them below. Please note that those are major concerns in my evaluation of the submission.
> >
> > - MEPOL: The algorithm presented in Mutti et al 2021 seems to address essentially the same problem of Hazan et al 2019, Liu & Abbeel 2021, and most of the subsequent works in this stream (some references are provided above) including the work the authors are submitting here. There are slight differences in how the objective is formulated, but not in the nature of the problem. Hazan et al 2019 and this submission consider the entropy of the discounted state distribution $H ((1 - \gamma) \sum_{t = 0}^\infty \gamma^t d_t^\pi (s))$, which is the close relative of the finite-horizon state distribution $H ( \frac{1}{T} \sum_{t = 0}^T d_t^\pi (s))$ (note that $t$ is the index of the step of an episode in both the formulation). Liu & Abbeel 2021 actually make a slight departure, as they seek to maximize the entropy in the replay buffer instead of $d^\pi$. Despite this and various other implementation differences, the method by Liu & Abbeel is definitely built over the ideas from Mutti et al.
> >
> > - APT uniform mixture: Again, APT by Liu & Abbeel does not seem to train a uniform mixture of policies, but to train a single policy (the last-iterate policy to be used as initial policy for the fine-tuning) with some degree of off-policy data. These two things are significantly different.
> >
> > - "Tuning $\eta$ is unnecessary in MaxEnt-H": I would reiterate my concern above. The paper should demonstrate that similar convergence guarantees can be obtained with a uniform mixture to support this claim. Otherwise, the authors can instead drop the claim and just mention that tuning $\eta$ does not lead to improvement in their experiments.
> >
> > - Theorem 1: "sub-optimality of a policy" means "performance gap between a policy and the global optimal policy". Thus, from the statement of Theorem 1, $\varepsilon$ appears to be an upper bound on the sub-optimality. What I meant with my comment is that proving a smaller $\varepsilon$ does not imply that the *true* sub-optimality is smaller.
> >
> > - Boundedness: The kNN estimator in eq. 6 is not bounded in general.
> >
> > - Experimental significance: While 4 seeds is better than 1, those plots still have little statistical meaning without a clear description on how the plots are obtained (please refer to the references provided in the review).
> >
> > - Discounted setting: In the experiments, are the episodes generated from a discounted MDP, i.e., with a probability of an episode ending at any step equal to $\gamma$? Otherwise, there may be a mismatch between the theoretical formulation and the experimental setting.
> >
> > - In the final comment, as well as the paper, the authors are dismissing previous works, such as Seo et al 2021, Yarats et al 2021, Mutti et al 2022 and others, raising bizarre motivations. Note that the work of other researchers, especially when it has been published at major conferences after a reviewing process, shall be taken into serious consideration.

---

> ### Author Response · Authors · 2024-11-20
>
> **Tuning $\eta$ and Theorem 1**
>
> Given kNN \textit{density} estimation, MaxEnt-V is algorithmically identical to MaxEnt-H with $\eta \rightarrow 0$ and a hot-starting initialization. Therefore, the convergence guarantees for a uniform mixture are already provided in Theorem 1 of our paper. The distinction is because that the main Theorem 4.1 of MaxEnt-H is based on the following settings: $\varepsilon_1 = 0.1\varepsilon$, $\varepsilon_0 = 0.1\beta^{-1}\varepsilon$, $\eta = 0.1\beta^{-1}\varepsilon$, and $T = \eta^{-1} \log \left(10B\varepsilon^{-1}\right)$. This setting is quite unrealistic, as we cannot directly control $\varepsilon_0$ or $\varepsilon_1$. If we eliminate these setting, Theorem 4.1 of MaxEnt-H would be in the similar expression Theorem 1 in our paper, with $\varepsilon = B e^{-T\eta} + 2\beta\varepsilon_0 + \varepsilon_1 + \eta\beta$ when $\eta \rightarrow 0$ (see lines 314 to 317).
>
> If we adopt a similar setting to "cheat," we can simply set $\varepsilon_1 = 0.1\varepsilon$, $\varepsilon_0 = 0.1 \beta^{-1}\varepsilon$, and $T = 1.43(B + \beta[\rho + \ln(T)]) \varepsilon^{-1}$. The corresponding guarantee, expressed in a similar form to Theorem 4.1, becomes $T > 1.43(B + \beta[\rho + \ln(T)]) \varepsilon^{-1}$, or equivalently $T/(\beta \ln T + \text{Constants}) > 1.43 \varepsilon^{-1}$.
>
> It is important to note that the goal of Theorem 1 is to compare regret and sample complexity. It represents a correction to Theorem 4.1 itself, incorporating a uniform mixture. Theorem 4.1 is inadequate for describing a uniform mixture, as it multiplies by $\frac{1}{\eta}$ during the derivation of $\varepsilon$.
>
> **Boundedness**
>
> Recall the definition of $B$-boundedness: $ \|\nabla R(X)\|_\infty \leq B $, where $R$ represents the particle (kNN) entropy and $X$ lies within the state space. In this case, $B$ definitely exists, and we can compute it using brute force based on all historical state distributions. However, given that we operate in a large continuous space with millions of state points, obtaining a large value of $B$ is particularly challenging.
>
> **Discounted setting**
>
> In the experiments, the episodes are generated from a discounted MDP, consistent with the approach used in MaxEnt-H.
>
> **Related works**
>
> We agree with the suggestion to include additional related works. We have already cited Seo et al. (2021), and we will also cite Yarats et al. (2021) as a similar method to Seo et al. (2021), with differences in the encoder architecture. Additionally, we will include Mutti et al. (2022), which discusses scenarios involving unique optimal actions.

---

> ### Author Response · Authors · 2024-11-20
>
> We appreciate your response but encourage you to carefully review the main theorem in MaxEnt-H and the primary reward function in MaxEnt-LA (you may find Seo et al., 2021 useful for its simplicity). Most of the questions raised concern algorithmic details related to these elements. It is important to note that, given the kNN \textit{density} estimation, MaxEnt-V is algorithmically identical to MaxEnt-H with $\eta \rightarrow 0$ and a hot-starting initialization. Similarly, MaxEnt-V is algorithmically equivalent to MaxEnt-LA when using a random encoder (Seo et al., 2021) and without the replay buffer freezing trick. These three methods can be transitioned into one another with at most five lines of code.
>
> **MEPOL**
>
> The statement "the slight difference does not change the nature of the problem" is neither formal nor correct. We can interpret $H\left(\frac{1}{T}\sum_{t=0}^{T}d_t^{\pi}\right)$ as a finite-horizon, discount-free ($\gamma \rightarrow 1$) version of MaxEnts’ objective, $H\left(\sum_{t=0}^{\infty}d_t^{\pi}\right)$, only if all induced trajectories have the same fixed length $T$. However, this assumption does not hold in environments such as Mujoco or in real-world scenarios, where stopping conditions (e.g., death, crash) are present. Specifically, if a trajectory $\tau$ terminates early at step $t = T(\tau) < T$, then $H\left(\frac{1}{T}\sum_{t=0}^{T}d_t^{\pi}\right)$ sums the probabilities over only the $T(\tau)$ steps in the trajectory, but still divides by the fixed $T$. This would incorrectly encourage the generation of longer trajectories. In fact, the finite-horizon version of MaxEnt should be expressed as $H\left(\sum_{t=0}^{T(\tau)}d_t^{\pi}\right)$, where $T(\tau) < T$.
>
> More importantly, the MaxEnt methods discussed in our paper \textbf{do not} share key algorithmic techniques with MEPOL. The fundamental components of MaxEnt-H, MaxEnt-LA, and MaxEnt-V are dense reward functions, $r(s)$, and our paper demonstrates that these reward functions are identical when kNN is used ($r(s) = \log(\|s - s^{k\text{NN}}\|_2^p)$) given by \textit{vanilla} kNN estimator. Each state receives a reward value compared to all historical states $D = [s_i]$. This simple reward function can be used in any RL algorithms. In contrast, MEPOL employs a trajectory-wise data buffer $D = [\tau^t_i, s_i]$, where $\tau^t_i $ the previous states within the same episode, and then a weight is computed to implicitly implement \textit{importance-weighted} (IW) kNN estimator.
> As a result, MEPOL cannot be directly implemented in other off-policy RL algorithms, such as SAC, without further modification.
>
> Therefore, we cite it in the related works section of our paper, rather than discussing it in detail in our paper.
>
> **APT uniform mixture**
>
> As we addressed in the previous comment, MaxEnt-LA trains a SAC with the non-stationary reward function for 1,000,000 steps. Although MaxEnt-LA reports only the final policy, i.e., $\pi_{1,000,000}$, we can still obtain the mixture of 1,000,000 policies from the training history by saving the SAC parameters. Recall the definition of a mixed policy in MaxEnt-H: "The policy $\pi_{mix}$ is one where, at the first timestep $t = 0$, we sample policy $\pi_i$ with probability $\alpha_i$, and then use this policy for all subsequent timesteps." The induced states of the mixed policy with uniform sampling can be obtained by sampling multiple policies from the 1,000,000 policies and executing them.
>
> In fact, this is unnecessary. The induced states of the mixture are theoretically equivalent to all the states visited during the training process.

---

> > ### Comment · Reviewer_YseA · 2024-11-20
> >
> > I thank the authors for the follow-up response. I feel I have now the elements I need to go into the private discussion with the reviewers. I am leaving some comments and suggestions below that the authors may use for future versions of the manuscript. There is no need of further response by the authors.
> >
> > - MEPOL: It is right that MEPOL is designed for a finite horizon setting. It is not necessary to discuss here whether discounted or finite-horizon formulations are more realistic/useful in practice (surely enough, one can make the case that the discounted setting is not realistic). What matters is that both finite-horizon and discounted settings are very common in the literature and arguably worth studying. Instead, I do not think it is acceptable to dismiss a paper as unrelated because it tackles the finite-horizon setting (anyway, MEPOL can be easily adapted to the discounted setting). That paper by Mutti et al. (2021) is relevant here because it was the first to use the entropy estimation technique that we later found in Liu & Abbeel (2021), Seo et al. (2021), Yarats et al., (2021) and many others (including this submission). This shall be credited in the manuscript.
> >
> > - APT: Of course one can treat the sequence of policy learned by APT as a uniform mixture, but that was not the intended design of the algorithm. Indeed MaxEnt-V presented in this paper closely resembles MaxEnt, but it is fairly different from APT.
> >
> > - If the Theorem 4.1 of Hazan et al. is incorrect, I suggest to mention it clearly in the paper with supporting derivations. Especially considering that Hazan et al. is a well-established paper in the literature.
> >
> > - Boundedness: It just gets to have $k$ identical states in the batch used to compute the reward (e.g., a deterministic initial state) to make the kNN radius zero and the reward unbounded.
> >
> >
> > Overall, I like the idea of freezing the intrinsic reward for multiple episodes of interactions. However, it is unclear whether the non-stationarity of the reward ever created a problem for previous methods (Liu & Abbeel, Seo et al., Yarats et al... They all present quite compelling empirical results) and the paper does not nearly enough to support the claim that freezing benefits performance. I believe the latter aspect is still worth studying and focusing the effort on that would make for a nice contribution. Instead, this takes a backseat in the current manuscript in favour of the unifying framework, which contribution to the literature is honestly questionable.

---

> > > ### Author Response · Authors · 2024-11-21
> > >
> > > We thank you for your response. Considering that the discussion will be mandatorily made public, we must address the factual errors.
> > >
> > > **MEPOL**
> > >
> > > **In our initial submission, we had already cited MEPOL and the works you mentioned in the manuscript**, including Mutti et al. (2021), Seo et al. (2021), and Yarats et al. (2021). Due to the significant algorithmic and theoretical differences, which we have discussed above, we positioned these works in the related works section without further discussion. Additionally, Mutti et al. (2021) is not the first to employ the kNN entropy estimation technique, which was already implemented in MaxEnt-H in 2019.
> > >
> > > **APT**
> > >
> > > This is not accurate. We have demonstrated that MaxEnt-H and MaxEnt-LA are essentially identical when appropriate hyperparameters are selected. Auxiliary modules, such as encoders, are not the focus of this paper, as they can also be integrated into MaxEnt-H.
> > >
> > > **Theorem 4.1**
> > >
> > > We will provide further clarification on lines 311 to 317.
> > >
> > > **Boundedness**
> > >
> > > Both MaxEnt-H and MaxEnt-LA introduce a scaling factor to avoid $-\infty$, i.e., $r(s) = \log(\|s - s^{k\text{NN}}\|_2^p + \delta)$. Further details can be found in their official implementations. We have kept it the same.

---

### Official Review · Reviewer_nXsJ · 2024-11-02

**Soundness:** 4
**Presentation:** 2
**Contribution:** 1
**Rating:** 3
**Confidence:** 3

**Summary:**

The paper first presents two established approaches to tackle the maximum state entropy maximization problem in reinforcement learning. Then, they first show how the two algorithmic schemes fundamentally aim to solve the same formal problem and present limitations of both schemes. Crucially, they identify that the policy sampling schedule is sub-optimal for one, while the policy update strategy is sub-optimal for the other one. Towards bridging the two schemes in a unified manner and overcoming these issues they propose a new algorithm, provide a theoretical analysis of its finite-time sub-optimality, and provide experimental comparisons with the previously mentioned existing algorithms.

**Strengths:**

CLARITY:
- The paper presents a clear and detailed comparison between two existing RL algorithms for state entropy maximization.
- The intentions of the paper are clearly specified from the abstract and followed along the paper in a clear and coherent manner.

QUALITY:
- After a brief check of the proof, the deviation from the original analysis of Hazan et al. of Theorem 1 (pag. 14 of the paper) seems correct and well explained.

ORIGINALITY:
- Although maybe common in other areas, I am not aware of other works in this context leveraging the Euler-Mascheroni constant to analyze the telescoping sum as done in the proof of Theorem 1.

SIGNIFICANCE:
- The problem tackled within the paper, namely state entropy maximization, is a fundamental problem for RL as it tackles from first principles the issue of exploration in RL. As a consequence, investigation in this direction is highly relevant for RL and beyond.
- Theorem 1 closes a gap between theory and practice in the choice of policy sampling schedule.

**Weaknesses:**

ORIGINALITY and SIGNIFICANCE:
Unfortunately, the paper seems to suffer a quite fundamental issue in terms of originality and significance because of the following points holding together:
1) The work fundamentally aims to 'build a unified perspective' on  the maximum state entropy problem by bridging two established schemes, namely the algorithms presented by Hazan et al. (MaxEnt, here renamed MaxEnt-H) and by Liu et al. (APT, here renamed MaxEnt-LA).
2) Crucially, the work by Liu et al. cites (and seems to build on) work [1]. This work fundamentally already presents the algorithmic ideas used in Liu et al. (namely the non-parametric entropy estimate to scale to non-tabular domains) in the context of the maximum state entropy problem presented by Hazan et al. (Sec. 4 of [1]). Moreover, it proposes an algorithm, named MEPOL, that seems to be nearly analogous to the one proposed by the authors (MaxEnt-Veritas) as a subcase. In particular, by choosing a high value of $\delta$ in MEPOL, it seems that the policy update scheme corresponds to the one in MaxEnt-Veritas (as in Hazan et al.), while using the policy sampling scheme as in Liu et al.
3) The authors cite [1] both in the Introduction and Related Works section, where they claim that [1] 'focuses on maximizing trajectory-wise state entropy' which is a 'fundamentally different objective'. Although this is the case for later works of the same author that alike the mentioned work by Jain et al. optimize trajectory-wise state entropy, this does not seem to be the case in [1], where the notion of entropy in MEPOL is not trajectory-wise.

As a consequence, it seems to me that the authors aimed to bridge two works that were already deeply (historically and formally) connected by a misinterpreted well-established reference. Although the submitted paper brings a new theoretical result (Theorem 1) by a slight modification of the analysis in Hazan et al., this seems very limited in terms of contribution and novelty compared with what the authors claim (e.g., in the abstract), which was arguably already achieved in large part.

CLARITY and QUALITY:
- I believe that the first 5 pages of the paper can be significantly sharpened in their presentation, which currently seems loose and slightly hard to follow at points.
- I would suggest to present Propositions 2 and 3 not as propositions as they seem simple calculations based on existing theorems and could be integrated within the text to improve the flow of the paper.


[1] Mirco Mutti, Lorenzo Pratissoli, and Marcello Restelli. Task-agnostic exploration via policy gradient of a non-parametric state entropy estimate.

**Questions:**

Did I misinterpret something within points 1-3 above that renders the conclusion fundamentally wrong?

---

> ### Author Response · Authors · 2024-11-14
>
> We appreciate your detailed response, but we humbly disagree that the contribution of this paper is limited. This paper is the first to bridge two of the most important tracks on maximum state entropy (MaxEnt), helping to clarify the disorganized literature within this community. This work help prevent further misinterpretations, such as those made by the reviewer, in fundamentally misreading previous works including MaxEnt-H, MaxEnt-LA, and MEPOL [1], which have led to conclusions based on factual errors. We elaborate on these points in detail as follows.
>
> **Weaknesses:**
>
> We believe you misinterpret the relationship between MEPOL and MaxEnts discussed in this paper. MEPOL is a particularly misleading work for readers seeking to understand MaxEnt, as it claims to improve MaxEnt-H by introducing kNN. Works like MaxEnt-LA cite it, but in fact, these works only reference MEPOL in passing, acknowledging its early connection to kNN, without ever directly comparing it or discussing it in detail. This has led some to assume a strong connection, which is, in fact, incorrect. MEPOL is **NOT** premilitary to MaxEnt-LA.
>
> From a motivational perspective, MEPOL seeks to maximize the entropy of the average state distribution $H\left(\frac{1}{T}\sum_{t=1}^{T}d_t^{\pi}\right)$, defined over episodes (note the index $t$ within an episode), while MaxEnt aims to directly maximize the state distribution $H(d^{\pi})$. Theoretically, MEPOL reduces the general MaxEnt problem to a finite-horizon setting, which requires a fixed hyperparameter (episode length) $T$ for each experiment. Consequently, MEPOL has to remove the default termination rules for safe learning in Mujoco to maintain a fixed horizon length, which leads to illegal poses. In contrast, MaxEnt methods allow for varying episode lengths in each run.
>
> We can interpret $H\left(\frac{1}{T}\sum_{t=0}^{T}d_t^{\pi}\right)$ as a finite-horizon, discount-free ($\gamma \rightarrow 1$) version of MaxEnts’ objective, $H\left(\sum_{t=0}^{\infty}d_t^{\pi}\right)$, only if all induced trajectories have the same fixed length $T$. However, this assumption does not hold in environments such as Mujoco or in real-world scenarios, where stopping conditions (e.g., death, crash) are present. Specifically, if a trajectory $\tau$ terminates early at step $t = T(\tau) < T$, then $H\left(\frac{1}{T}\sum_{t=0}^{T}d_t^{\pi}\right)$ sums the probabilities over only the $T(\tau)$ steps in the trajectory, but still divides by the fixed $T$. This would incorrectly encourage the generation of longer trajectories. In real worlds, the finite-horizon version of MaxEnt should be expressed as $H\left(\sum_{t=0}^{T(\tau)}d_t^{\pi}\right)$, where $T(\tau) < T$.
>
> From an information-theoretic perspective, MEPOL establishes a projection from the state space to a space defined by the average probabilities within trajectories. This differs significantly from state entropy $H(d(s))$. The design of such projections generally results in fundamentally different information-theoretic metrics and properties. Further details on these information-theoretic definitions can be found in the [2, 3].
>
>
>
> From an implementation perspective, MEPOL fundamentally does not present the algorithmic ideas used in Liu et al. or even MaxEnt-H. The fundamental components of MaxEnt-H, MaxEnt-LA, and MaxEnt-V are dense reward functions, $r(s)$, and our paper demonstrates that these reward functions are identical when **vanilla** kNN density estimation is used ($r(s) = \log(\|s - s^{k\text{NN}}\|_2^p)$). Each state receives a reward value based on all historical states $D =[s_i]$. This simple reward function can be used in any RL algorithms. In contrast, MEPOL employs a trajectory-wise data buffer $D = [\tau^t_i, s_i]$, where $\tau^t_i $ denotes the previous states within the same episode, and then a weight is computed to implicitly implement **importance-weighted** (IW) kNN estimator, i.e., $H(\bar{d}^{\pi}_T|\bar{d}^{\pi’}_T)$, where $\bar{d}_T = \frac{1}{T}\sum_0^Td_t^{\pi}$.
> As a result, MEPOL cannot be directly implemented in other off-policy RL algorithms, such as SAC, without further modification.
>
> Therefore, we menioned it in the related work section without further discussion.
>
>
> CLARITY and QUALITY:
> 1. We recommend that the reviewer set aside any prior misconceptions about MEPOL and read MaxEnt-H, MaxEnt-LA, and our paper in sequence. We believe some misunderstandings arise from the confusing notions in previous works. For instance, in the symbols $i$, $t$ and $T$  We have reorganized these elements for clarity.
> 2. We will present them in a concise manner.
>
> Crucially, we do not believe it is appropriate to reject a paper solely because its claims and methods are perceived as too simple or lacking complex modifications, regardless of its correctness. The Heliocentrism, for instance, should not be dismissed simply because it is conceptually simpler or only slightly modifies the center while removing epicycles from the Geocentrism.

---

> > ### Comment · Reviewer_nXsJ · 2024-11-22
> >
> > As mentioned by the authors there are two main ways to look at state-entropy maximization. Either by evaluating the entropy of the distribution induced by aggregating state visitations from different trajectories, or in a trajectory-wise manner that aims to measure diversity within a single trajectory. Mathematically, the difference in the objectives lies in whether one optimizes for the average entropy over the distribution induced by multiple trajectories or for the entropy of the average distribution induced by multiple trajectories.
> >
> > MEPOL, alike the algorithm presented by the authors, MaxEnt-H and MaxEnt-LA tackles the former case, where one wishes to optimize the entropy of a distribution induced by multiple trajectories. Although MEPOL tackles a finite-horizon setting, it does not seem to be trajectory-wise and therefore its objective seems nearly identical to the one considered by the authors.
> >
> > Clearly, the authors present a slightly different setting than the one for which MEPOL has been developed (namely finite horizon), but currently it is not clear to me why one could not extend MEPOL to work in this setting as well. More specifically, since this work aims to 'build a unified perspective' this aspect is quite fundamental and is not discussed at all within the paper, where, on the contrary, MEPOL is treated as if its objective is a fundamentally different one. I believe that neglecting this relation would only decrease clarity within the literature.

---

> ### Author Response · Authors · 2024-11-14
>
> [1] Mirco Mutti, Lorenzo Pratissoli, and Marcello Restelli. Task-agnostic exploration via policy gradient of a non-parametric state entropy estimate.
>
> [2] Alfréd Rényi. On measures of entropy and information. 1961.
>
> [3] Principe, Jose C. Information theoretic learning: Renyi's entropy and kernel perspectives. Springer Science & Business Media, 2010.

---

### Official Review · Reviewer_nyAP · 2024-11-03

**Soundness:** 3
**Presentation:** 4
**Contribution:** 3
**Rating:** 8
**Confidence:** 3

**Summary:**

The authors study the two algorithms by Hazan (MaxEnt-H) and by Liu \& Abbel (MaxEnt-LA), which both compute a mixture of policies to maximize the (discounted) state entropy of an agent evolving in an infinite-time MDP. A priori, the reward function they maximize is different, the weights of the mixture are different, and the number of policy updates are different before updating the intrinsic reward estimate. The authors nevertheless prove that (1) the intrinsic rewards both algorithms optimize are proportional to each other when computed using kNN, (2) the parameter $\eta$ for computing the mixture in MaxEnt-H is unnecessary, and a uniform mixture as in MaxEnt-LA is sufficient (3) it is better to completely optimize the policy before updating the estimate of the intrinsic reward function, as advocated by Hazan in MaxEnt-H. Based on these three observations, the authors introduce a new algorithm (MaxEnt-V) by combining the methods from Hazan and Liu \& Abbel. They remove the unnecessary steps from Hazan, and compute the mixture as in Liu \& Abbel, but optimize the policy as in Hazan before updating the intrinsic reward estimate. The algorithm is $\epsilon$-optimal, meaning that the agent is at most suboptimal by $\epsilon$, and under some assumptions $\epsilon$ decreases by $(B + \beta \ln T) / T$ where $T$ is the number of iterations and where $B$ and $\beta$ are constant. In practice it performs at least as well as the algorithms from Hazan and from Liu \& Abbel.

Warning: I must apologize beforehand for not having checked the demonstrations in the appendix.

**Strengths:**

1. The paper is well written and easy to follow.
2. The problem addressed is interesting to the community.
3. The algorithm the authors propose is well motivated, with theoretical guarantees, and tested on several problems, where it also outperforms the alternative methods.

**Weaknesses:**

1. I would appreciate if the authors could clarify if their algorithm is always to be favoured compared to that of Hazan and Liu \& Abbel, or if there may be configurations in which their method would fail and where the others would not.
2. Paragraph line 313 to line 317 is pretty unclear to me. Could the authors explicitly provide the order of $\epsilon$ when $\eta \rightarrow 0$. Also, the bound of Hazan seems to be exponentially decreasing in T, wich a priori looks better than the bound of MaxEnt-V.
3. In Figure 4, results are represented in term of epochs. To my understanding, an epoch is a step $t$ in algorithm 1. Then, as in MaxEnt-H and MaxEnt-V, and epoch fully optimizes the policy, I suppose it is also much longer in terms of wall-time, compared to MaxEnt-LA. How do the figures look like as a function of the wall-time?
4. I do not agree with line 486, stating that the algorithms maximizing the action entropy lack the capability of exploring in absence of extrinsic reward. To me, without extrinsic rewards, these methods aim to explore the action space uniformly, which does not guarantee uniform state exploration. In opposition MaxEnt-H, MaxEnt-LA, and MaxEnt-V aim to explore the state space uniformly, which does not guarantee action space exploration. Both approaches have different intrinsic motivation (i.e., exploration objective), and it is possible to construct examples for which uniform action exploration outperforms uniform state exploration, and vice versa.
5. In paragraph line 508, there is an important distinction to make between parametric methods. Some are explicitly based on the entropy. The intrinsic motivation is to maximize the entropy of some distribution, which is typically approximated with a neural density estimator. Other methods are based on the uncertainty of some model. The intrinsic motivation is to take actions for which a parametric model over states and/or rewards provides different outcomes compared to the MDP realization. The distinction is particularly important in the current work as the first class of algorithms optimizes the same objective as the method the authors presented (and may have been added in the experiments). I think in particular that the related work should include [1, 2, 3], and probably other, more recent, works.

[1] Lee, L., Eysenbach, B., Parisotto, E., Xing, E., Levine, S., & Salakhutdinov, R. (2019). Efficient exploration via state marginal matching. arXiv preprint arXiv:1906.05274.

[2] Guo, Z. D., Azar, M. G., Saade, A., Thakoor, S., Piot, B., Pires, B. A., ... & Munos, R. (2021). Geometric entropic exploration. arXiv preprint arXiv:2101.02055.

[3] Islam, R., Ahmed, Z., & Precup, D. (2019). Marginalized state distribution entropy regularization in policy optimization. arXiv preprint arXiv:1912.05128.

**Questions:**

1. I cannot see the reward functions in the three algorithms that are stationary and those that are not. In particular the sentence line 183 confuses me.
2. Line 195 to line 199, might their be a confusion between MaxEnt-V and MaxEnt-H?
3. Line 488, I suppose the entropy function is concave and not convex.

---

> ### Author Response · Authors · 2024-11-19
>
> We appreciate your valuable response. We address your concerns as follow.
>
> **Weakness**
>
> 1.	Compared to MaxEnt-H in non-tabular experiments, our method should consistently perform better. We have shown that $\eta$ should always be very small, and more critically, the sampling probability $\alpha_t = \eta(1-\eta)^{t-1}$, with a fixed $\eta$, can never satisfy the condition $\sum_{t=0}^T \alpha_t = 1$.
> Compared to MaxEnt-LA, the performance of MaxEnt-V will be outperformed if the reward function $r_t$ changes slowly as $t$ increases. In this case, MaxEnt-LA can be seen as maximizing an approximately stationary reward function over recent steps. Consequently, there is no need to freeze the replay buffer as in MaxEnt-V, since doing so reduces sample efficiency.
> This situation typically arises in environments with small state spaces, such as small mazes or simple graphs. Recall that we adopt kNN (particle) entropy. Since the nearest neighbors of visited states do not change significantly, $r_t$ will become an approximately stationary reward function soon.
>
> 2.	Apologies for the confusion. The value of $\varepsilon$ as $\eta \rightarrow 0$ is exactly given by Eq. (12) in Theorem 1 of our paper. The only distinction is that we specifically aim to maximize kNN (particle) entropy, eliminating the need to discuss $B$ and $\beta$.
> MaxEnt-H fails to give Eq. (12) due to the multiplication by $\frac{1}{\eta}$ in the derivation of $\varepsilon$. Consequently, the $\varepsilon$ obtained by MaxEnt-H, i.e., $\varepsilon = B e^{-T\eta} + 2\beta\varepsilon_0 + \varepsilon_1 + \eta\beta$, cannot be used to describe the scenario as $\eta \rightarrow 0$.
>
>
> 3.	Apologies for the confusion. In each epoch (indexed by $t$ in the MaxEnt-V and MaxEnt-H algorithms), we train MaxEnt-H and MaxEnt-V for 500,000 steps. For MaxEnt-LA, the total number of training steps is set to $10,000,000$, ensuring that the SACs are updated an equal number of steps. Each epoch indicates $500,000$ steps.
> The states in Figs. 4 are obtained as follows: For MaxEnt-V and MaxEnt-H, we sample $\pi$ from the set of mixed policies $\pi_1, \pi_2, \dots, \pi_t$ to induce 100,000 states. An epoch corresponds to the update of the reward function $r_t$ for MaxEnt-H and MaxEnt-V, along with the corresponding training steps for SAC in MaxEnt-LA. For MaxEnt-LA, there would be millions of steps and corresponding policies in the set, as $r_t$ is updated in each step. Therefore, we can simply sample 100,000 states from all historical states to obtain the induced states across all historical policies. This approach is theoretically equivalent to recording all policies and executing them uniformly.
>
>
> 4.	We will improve this expression, as you are correct. SACs can also explore by setting the rewards to be consistently zero. SACs aim to maximize $H(p(a|s))$, while MaxEnt aims to maximize $H(p(s))$. The key difference between the two approaches lies in the spaces they seek to explore: the state space for MaxEnt and the action space for SAC.
>
> 5.	Thank you for your recommendations regarding the related works; we will certainly improve this section. More specifically, [1] is highly related to MaxEnt-LA. The key difference is that [1] minimizes the KL divergence to a prior uniform distribution $\rho$, which requires knowledge of the state space. In contrast, [3] provides an implicit variational approximation of the state distribution, whereas MaxEnt-LA uses a non-parametric kNN approach. [2] differs in that it aims to maximize the entropy of the average state distribution, $H\left(\frac{1}{T}\sum_{t=1}^{T}d_t^{\pi}\right)$, rather than $H(d(s))$. This formulation reduces the general MaxEnt problem to a finite-horizon setting, which necessitates a fixed hyperparameter (episode length) $T$, similar to MEPOL [4]. Therefore, it should be considered as a trajectory-wise entropy maximization.

---

> > ### Comment · Reviewer_nyAP · 2024-11-26
> > **Response to comment**
> >
> > Thank you for responding, I appreciated reading your paper too.

---

### Official Review · Reviewer_MFum · 2024-11-05

**Soundness:** 3
**Presentation:** 3
**Contribution:** 3
**Rating:** 5
**Confidence:** 3

**Summary:**

This paper provides a stronger proof for learning a policy that achieves maximum state-entropy, by using a simpler and more adaptive policy-sampling strategy. It further claims to unify two popular approaches to maximum-entropy RL. Empirical results suggest that this improved sampling strategy leads to more exploratory and uniform policies.

**Strengths:**

The authors do a good job of explaining the shortcomings of the bounds provided by Hazan. The thorough description of why small η leads to unrealistic T was very helpful for intuition. The proof result, if I understand it, is a significant improvement in that regard. I generally thought the mathematical writing in this paper was strong.

**Weaknesses:**

I understand this is largely a theoretical paper, but nevertheless the experimental section of this paper is not very well-described or thorough. That’s the main reason for my “weak reject” – if the authors sufficiently address these concerns, I’ll gladly raise it.

From the plots, it appears that the experiments are run with one seed – if so, this isn’t acceptable, but can be easily rectified.

I’m very confused how you are using the NGU bonus, which is history-dependent, while not using a memory-based (e.g. LSTM) policy. Can you explain? The NGU bonus as I understand it doesn’t make much sense without recurrence. I also found it a bit unfair to criticize MaxEnt-LA for using a non-stationary reward when you use one here (of almost the same form, even).

Many details are missing for experiment configuration as well. How much training data is gathered each epoch? How much training is done per epoch? Was there a concrete measurement for the “epsilon-accurate stopping criteria” in the empirical results, or was it based on the number of training steps (it's hard for me to understand what "train to convergence" means in the high-dimensional setting)? What does an “epoch” mean for MaxEnt-LA? What’s the scale of the NGU bonus? Are the projection functions consistent across methods (the random projections for each method need to have the same parameterization for the comparison to be fair)?

How can there be a slight downwards tick for the blue line for Ant for “total unique states visited during training”?

To clarify, in the appendix when you describe projecting down the states to a 7-dimensional space, that’s only for entropy calculation, correct? And not projecting down for control as well?

I would also encourage the authors to choose a different name for their method. “MaxEnt-Veritas” seems to imply that the authors view their method as the “one final and true MaxEnt” method, while also implying that there was something false about prior work. The main difference between your method and Hazan is an improved policy-selection scheme, and so I don’t think this method is any more “truthful” than theirs, possibly just more efficient. And I’m sure someone (maybe even you) will at some point improve upon this method as well. I similarly think the title is not as informative about the methodology as it could be.

**Questions:**

Largely the empirical questions above, which is what my review currently hinges on. In brief:

* Can you run with more seeds?
* Can you include much more information on experimental configuration in your paper? Especially, can you describe what an "epoch" means, and if epsilon-accurate stopping is used, how that works in practice? Can you clarify whether projection is only for entropy computation or control as well? I should in theory be able to recreate your experimental protocol from your paper, but at the moment that's the not the case.
* As I understand, the NGU bonus as applied in the original paper (computed using previous states seen throughout a given trajectory) seems incorrect for a non-recurrent method. Would you share your thoughts on this? And can you comment on the non-stationarity introduced by NGU in relationship to the criticisms of MaxEnt-LA?
* Can you explain the slight downward tick in the blue line, since that doesn't seem possible given the plot's description?

---

> ### Author Response · Authors · 2024-11-19
>
> We appreciate your valuable response. It is our oversight that we did not pay sufficient attention to the detailed description in experiments. We have added more details to address your concerns, as outlined below.
>
> **Experimental details and seeds**
>
> The curves we provide represent the average scores from 4 runs. In the revised PDF, we have updated the curves along with the corresponding variances. In each epoch (indexed by $t$ in the MaxEnt-V and MaxEnt-H algorithms), we train MaxEnt-H and MaxEnt-V for 500,000 steps. These numbers were selected because they are sufficiently large to achieve a small $\varepsilon_1$, rather than manually setting an $\varepsilon_1$-accurate stopping criterion. For MaxEnt-LA, the total number of training steps is set to $10,000,000$, ensuring that the SACs are updated an equal number of times.
>
> The states in Figs. 4 and 5 are obtained as follows: For MaxEnt-V and MaxEnt-H, we sample $\pi$ from the set of mixed policies $\pi_1, \pi_2, \dots, \pi_t$ to induce 100,000 states. An epoch corresponds to the update of the reward function $r_t$ for MaxEnt-H and MaxEnt-V, along with the corresponding training steps for SAC in MaxEnt-LA.
>
> For MaxEnt-LA, there would be millions of steps and corresponding policies in the set, as $r_t$ is updated in each step. Therefore, we can simply sample 100,000 states from all historical states to obtain the induced states across all historical policies. This approach is theoretically equivalent to recording all policies and executing them uniformly. The projection functions used for entropy calculation are consistent across methods and remain the same for all methods within each environment. (If these functions were different, the entropy values would vary significantly even for the same states.) The projection is used solely for entropy calculation purposes.
>
> **About NGU**
>
> The NGU can be considered an intrinsic reward function, $r_{ngu}$, designed to help avoid local optima in task-driven rewards, $r_{task}$. It is an auxiliary neural network irrelevant to the SAC and outputs a single reward value based on historical episodes. NGU requires no recurrence, as its episodic memory is not autoregressive.
>
> To implement NGU, users can simply add the outputs of the NGU to the regular task-driven rewards without modifying the SAC. The combined reward is given by $r_{task} + \alpha r_{ngu}$, where $\alpha$ is a hyperparameter. SAC is then trained using these combined rewards, which results in larger $r_{task}$ values. In our experiments, we treat $r^{LA}$ and $r^{H}$ as $r_{task}$ and add $r_{ngu}$ in Step 3 of the algorithms. The objective of NGU is to increase the values of $r_{LA}$ and $r_{H}$ after convergence, without considering the value of $r_{ngu}$.
>
> In our experiments, we found that this trick leads to larger $r_{LA}$ and $r_{H}$ values for all three methods, including MaxEnt-LA, Therefore, we implement this with $\alpha = 0.5$.
>
> **Downwards Tick**
>
> As discussed in the first point, we do not record all historical states in the entire training process for MaxEnt-V. What we did is to save policies (SACs) themselves using checkpoints, and then execute these policies to induce states to obtain figures and curves. Therefore, the unique state visited is not accumulated since we resample induced states after each epoch.
>
> **Something More**
>
> We greatly appreciate that you have read our paper in detail. We may adopt the name "Unified MaxEnt" for the framework.
>
> Historically, informal descriptions and misinterpretations, such as "MaxEnt-H is just a theoretical tool, while MaxEnt-LA belongs to a stream of practical methods," have been prevalent. We are the first to provide a theoretical and algorithmic blueprint that connects most of the influential works in maximum state entropy.
>
> This paper is not aimed at introducing tricks to improve performance by 4 or 5 % on a specific dataset. The experimental results are both predictable and explainable, as outlined in Theorem 1.

---

> > ### Comment · Reviewer_MFum · 2024-11-21
> > **Still unclear on NGU bonus**
> >
> > Hi, thanks for your response.
> >
> > I understand the NGU paper and intrinsic rewards fairly well. In NGU, at timestep $t$ the episodic bonus $r_{NGU}(s_t)$ does a KNN search over the $t$ states visited earlier in that episode. After each timestep, it's doing the search over a different set of states, so the reward is non-stationary. The point of this is to encourage intra-episode diversity. This wouldn't make sense without a recurrent Q-function, because a feed-forward Q-function could not model different values for the same state depending on when in a trajectory it was visited. You don't need recurrence to compute this bonus, but you do need recurrence for it to in any interesting way serve its intended purpose.
> >
> > Can you clarify what set of states the NGU bonus is computed using? Is it the same set of states used for the KNN entropy bonus? Or the episodic prefix for a given state? Please clarify this in the paper. Also, please include the value for $k$ used for KNN, both for NGU-bonus and the MaxEnt reward (and it seems Eq 6 is missing a summation over "k").
> >
> > I didn't catch this before, but referring to the KNN bonus as the "never-give-up regularizer" is confusing. The bonus you use from that paper is an episodic intrinsic reward, not a regularizer. In this paper as well it's not clear how the bonus is a regularizer, any more than the entropy reward is a regularizer (given that the two have extremely similar forms).

---

> ### Author Response · Authors · 2024-11-22
>
> We appreciate your valuable response. We have added more details to address your concerns, as outlined below.
>
> NGU
>
> If we understand correctly, you may suggest that the episode-based NGU reward should only be effective when using RL algorithms with recurrent value functions, such as Recurrent Replay Distributed DQN, to maximize it. However, this is not strictly necessary, as a widely used technique involves optimizing memory-based non-stationary rewards directly with normal off-policy RL. MaxEnt-LA itself serves as an example: Seo et al. (2021) maximize the non-stationary $r^{LA}$ directly using non-recurrent A2C and RAD.
>
> The reason why it works in episodic settings can be found in [1,2]. In brief, the reward is automatically transferred to a dense non-recurrent one via an implicit Monte-Carlo estimation when we sample a batch from the replay buffer to update agent. The non-recurrent agent would simply be encouraged to discover states within episodes that contain diverse states.
>
> Therefore, what we did is straightforward: we compute $r_{ngu}$ using Eq. (2) from the NGU paper in state space, setting $\alpha_t$ in Eq. (1) of the NGU paper to a constant value of 1, since $r^{LA}$ already serves as the long-term novelty detection bonus. Then, $r_{ngu}$ is added directly to $r^{LA}$, allowing us to obtain a larger $r^{LA}$ in each iteration. The primary objective remains to maximize $r^{LA}$.  We set $k$ to 3 for both $k$NN and use the same formulation as Eq. (6) from Seo et al. (2021). The $||\dots||_2$ denotes the norm for all $k$NN distances of s.
>
> Apologies for the confusion. It is more accurate to say that NGU serves as the intrinsic reward for another intrinsic reward, $r^{LA}$ (our goal). It was our oversight that we did not give sufficient attention to this, as we considered step 4 as "(approximately) computing the optimal $\pi$ to maximize $r^{LA}_t$", similar to the approach in MaxEnt-H.
>
> [1] Y. Efroni, N. Merlis, and S. Mannor, “Reinforcement learning with trajectory feedback,” in Proceedings of the AAAI conference on artificial intelligence
>
> [2] Z. Ren, R. Guo, Y. Zhou, and J. Peng, “Learning long-term reward redistribution via randomized return decomposition,

---

### Meta-Review · Area_Chair_DGwq · 2024-12-21

**Metareview:**

This paper proposes a new algorithm for maximum state entropy exploration, MaxEnt-V, which combines ideas from two existing methods, MaxEnt-H and MaxEnt-LA. The paper presents a unifying framework and demonstrates the effectiveness of freezing intrinsic rewards across multiple episodes.

However, the reviewers raise significant concerns. They note that the paper does not sufficiently credit key prior works, such as Mutti et al. (2021), and misrepresents certain aspects of APT. The empirical results lack statistical rigor, with unclear experimental details and a need for comparison with official implementations of prior methods. Theoretical concerns also arise, particularly regarding the assumptions in Theorem 1 and the unboundedness of kNN estimators. While the freezing intrinsic reward mechanism shows promise, the paper overemphasizes the unifying framework, which does not add enough novelty.

Overall, the paper would benefit from a clearer acknowledgment of prior work, stronger empirical validation, and a more focused exploration of the reward-freezing idea. With these revisions, it could make a valuable contribution to the field.

**Additional Comments On Reviewer Discussion:**

During the rebuttal period, the authors addressed several concerns raised by the reviewers, but key issues remained unresolved. The main criticism was the misleading claim of unifying two related algorithms, MaxEnt-H and MaxEnt-LA. Reviewers felt this claim was overstated, as MaxEnt-H is a theoretical framework and MaxEnt-LA is a practical implementation of similar ideas. The authors clarified their intentions but did not convincingly support the unification claim, which remained a central issue.

The theoretical support for freezing the reward was also questioned. While the authors argued it could offer empirical benefits, the theoretical foundation was weak, and the experiments did not sufficiently demonstrate its effectiveness. Reviewers also found the empirical results lacking statistical rigor, with issues in baseline comparisons and experiment design. Additionally, the presentation of related work was criticized for not fully crediting prior contributions, particularly the use of kNN entropy estimation.

Despite some interesting ideas, the paper failed to clearly establish its contribution, particularly in terms of unification and theoretical support. The empirical validation was insufficient, and the paper’s framing was misleading, leading to the final recommendation of rejection.

---

### Decision · Program_Chairs · 2025-01-22

Reject